# Prediction of Poisson's ratio for a petroleum engineering application: Machine learning methods

Fahd Saeed Alakbari[1]*, Syed Mohammad Mahmood[2,3], Mohammed Abdalla Ayoub[4]*, Muhammad Jawad Khan[3,5], Funsho Afolabi[3,5], Mysara Eissa Mohyaldinn[2,3], Ali Samer Muhsan[6]

1 Centre of Advanced Process Safety (CAPS), Universiti Teknologi PETRONAS, Seri Iskandar, Perak, Malaysia, 2 Center of Flow Assurance, Institute of Subsurface Resources, Universiti Teknologi PETRONAS, Seri Iskandar, Perak Darul Ridzuan, Malaysia, 3 Department of Petroleum Engineering, Universiti Teknologi PETRONAS, Bandar Seri Iskandar, Perak, Malaysia, 4 Department of Chemical & Petroleum Engineering, United Arab Emirates University, Al Ain, United Arab Emirates, 5 Institute of Hydrocarbon Recovery, Universiti Teknologi PETRONAS, Bandar Seri Iskandar, Perak, Malaysia, 6 Mechanical Engineering Department, Universiti Teknologi PETRONAS, Bandar Seri Iskandar, Perak, Malaysia

* ma.ayoub@uaeu.ac.ae (MAA); fahd.akbari@utp.edu.my, alakbarifahd@gmail.com (FAS)

## Abstract

Static Poisson's ratio ($v_s$) is an essential property used in petroleum calculations, namely fracture pressure (FP). The $v_s$ is often determined in the laboratory; however, due to time and cost constraints, quicker and cheaper alternatives are sought, such as data-driven models. However, existing methods lack the accuracy needed for critical applications, necessitating the need to explore more accurate methods. In addition, the previous studies used limited datasets and they do not show the relationships between the inputs and output. Therefore, this study developed a reliable model to predict the $v_s$ accurately using the nineteen most common learning methods. The proposed models were created based on a large data of 1691 datasets from different countries. The best-performing model of the nineteen models was selected and further enhanced using various approaches such as trend analysis to improve the model's performance and robustness as some models show high accuracy but show incorrect relationships between the inputs and output because the machine learning model only built based on the data and do not consider the physical behavior of the model. The proposed Gaussian process regression (GPR) model was also compared with published models. After the proposed GPR model was developed, the FP was determined based on the proposed GPR $v_s$ model and the previous $v_s$ models to evaluate their accuracy on the FP determinations. The best approach out of the published and proposed methods was GPR with a coefficient of determination ($R^2$) and average-absolute-percentage-relative-error (AAPRE) of 0.95 and 2.73%. The GPR model showed proper trends for all inputs. The cross-plotting and group error analyses also confirmed that the proposed GPR approach had high precision and surpassed other methods within all practical ranges. The GPR model decreased the residual error of FP from 87% to 26%. It is believed that such a significant improvement in the accuracy of the GPR model will have a significant effect on realistic FP determination.

**Data availability statement:** All relevant data are within the manuscript and its Supporting Information files.

**Funding:** Yayasan Universiti Teknologi PETRONAS (YUTP) (Cost Centre: 015LC0-451). The funders had no role in study design, data collection and analysis, decision to publish, or preparation of the manuscript.

**Competing interests:** The authors have declared that no competing interests exist.

**Abbreviations:** ν, Poisson's ratio; $v_s$, Static Poisson's ratio; $v_{dyn}$, dynamic ν; FP, fracture pressure; GP, Gaussian process; GPR, Gaussian process regression; $R^2$, coefficient of determination; AAPRE, average-absolute-percentage-relative-error; AAPRE, average-absolute-percentage-relative-error; DTc, compressional wave travel time; DTs, shear transit time; ANN, artificial neural network; RHOB, bulk formation density; UCS, compressive strength; FL, fuzzy logic; Vp, compressional wave velocity; ACE, alternating conditional expectation; Vs, shear wave velocity; FN, fuzzy neural; RF, random forest; CNN, convolutional neural network; LSTM, long-term short-term memory network; TA, trend analysis; SEA, Statistical error analysis; GEA, group error analyses; BAW, box and whisker; Emin, minimum absolute percent relative error; Emax, maximum absolute percent relative error; RMSE, root mean square error; SD, standard-deviation; FG, fracture gradient; D, Depth; PP, pore-pressure gradient; OBG, overburden gradient; SVM, support vector machine; MSE, mean squared error; MSE, mean squared error; PB, physical behaviour; KW, Kruskal–Wallis; ANOVA, analysis of variance; p-values, probability values; *HO, null hypothesis; IQR, interquartile range; RE, residual errors.

# 1 Introduction

In petroleum engineering, Poisson's ratio (ν) serves as a vital parameter in various determinations. It can be used for defining the horizontal stresses involved in constructing a model of geomechanical earth [1]. By utilizing *v*, petroleum engineers can optimize and control sand production [1]. The value of *v* can be influenced by changes in rock properties and lithology, particularly the bulk density [2–7]. For instance, soft rocks typically exhibit *v* values ranging from 0.1 to 0.3, while medium rocks like sandstone tend to have ν values between 0.2 and 0.3. Hard rocks, on the other hand, typically possess ν values between 0.3 and 0.4 [8].

ν is determined by applying two methods. The first method is applying the dynamic method and is known as dynamic ν ($v_{dyn}$). The $v_{dyn}$ of a rock is stated by the below equation:

$$\nu_{dyn} = \frac{V_P^2 - 2V_S^2}{2\left(V_P^2 - V_S^2\right)} \tag{1}$$

Where: $V_P$ is the compressional velocity, km/s; $V_S$ is the shear velocity, km/s; and $v_{dyn}$ is the dynamic Poisson's ratio.

The second method utilized in laboratory measurements is a static approach, which involves determining the static ν ($v_s$). The $v_s$ can be found using the below equation [9]:

$$\nu_s = -\frac{\varepsilon_y}{\varepsilon_X} \tag{2}$$

Where:

$\varepsilon_y$: strain in the x-direction; $\varepsilon_X$: strain in the y-direction; $\upsilon_s$: static ν.

The $v_s$ provides insights into the reservoir's actual behavior [9]. Nonetheless, obtaining $v_s$ through laboratory measurements can be expensive and time-consuming [10,11]. Therefore, some models were created to predict the $v_s$. Kumar et al. [12] conducted a study where they collected a dataset of 83 samples to establish a correlation for determining the $v_s$. However, their correlation applies only to isotropic rocks [12]. On the other hand, Khandelwal et al. [13] examined 11 datasets comprising different rock types from India. Their study used P-wave velocity or DTc (compressional wave travel time) as input to determine the vs [13]. In another study by Ranjbar-Karami et al. [14], they derived the $v_s$ as a function of the $v_{dyn}$. Their correlation yielded 0.3 $R^2$ using a Fuzzy-Inference-System (FIS) for finding the $v_s$, resulting in an improved $R^2$ value of 0.983 [14]. Brandås et al. [15] built a correlation linking the $v_s$ to the shear transit time (DTs), with the $v_{dyn}$. Feng et al. [16] employed a linear approach for determining $v_s$ utilizing 18 samples and data from low permeability reservoirs in China. In a different study, Gowida et al. [17] utilized an artificial neural network (ANN) approach to predict $v_s$ by analyzing 692 datasets from Saudi Arabia fields. Alakbari et al. [18] summarized the previous models based on the variables employed, the dataset sizes, the range of data, and the accuracy achieved by the authors in the previous correlations utilized for predicting the $v_s$.

Machine learning techniques have been employed to find the $v_s$ [19]. Singh [20] predicted $v_s$ for various rock types using ANN and neuro-fuzzy methods. On the other hand, Nejati et al. [21] also showed the determination for the $v_s$. Shalabi et al. [22] utilized linear regression for determining the $v_s$ by considering rock hardness and unified compressive strength (UCS). Abdulraheem et al. [11] employed some machine learning methods such as ANN methods to find $v_s$ using 77 data points. For the training data, the ANN method achieved an average absolute percentage error (AAPE) of 5.42%, while the fuzzy logic (FL) model achieved an AAPE of 8.20%. For the testing dataset, the ANN model had an AAPE of 5.16%,

and the FL model had an AAPE of 7.65% [1]. Al-Anazi et al. [23] estimated the $\nu_s$ using different parameters such as compressional wave velocity (Vp) based on 602 data points. The model was developed using the Alternating Conditional Expectation (ACE) method and achieved an $R^2$ value of 0.994 [23]. Tariq et al. [24] employed an ANN to establish $\nu_s$ based on shear wave velocity (Vs) and Vp based on 550 points, achieving $R^2$ = 0.82. Elkatatny et al. [25] utilized an ANN for calculating the $\nu_s$ applying some parameters such as bulk density and based on 610 points. Tariq et al. [26] employed the fuzzy neural (FN) method to predict $\nu_s$ using inputs such as gamma-ray, bulk density, porosity, Vs, and Vp. The model achieved an $R^2$ value of 0.97 using 580 data points. Alakbari et al. [18] used the gated recurrent unit to find the $\nu_s$ using several parameters such as bulk density. They showed that their model has an AAPRE of 32.23%. In addition, they only applied one method to predict the $\nu_s$. Cai [27] applied a random forest (RF)-convolutional neural network (CNN)-long-term short-term memory network (LSTM) to predict the Poisson's ratio based on the data from the gas field in China. The accuracy of their model is 94%. Therefore, the previous models need to improve their accuracy and explore other methods to determine the $\nu_s$. In addition, most of the previous studies used limited and specific datasets to determine the $\nu_s$ and they do not show the effects of inputs on the output or the relationships between the inputs and output. Therefore, this study developed a reliable model to predict the $\nu_s$ accurately using the Gaussian process regression (GPR). The proposed GPR model was created based on a wide range of datasets to make the model can be used for different datasets from different places. The proposed GPR model was also developed with the trend analysis to show the relationships between the inputs and output to prove the reliable model to find the $\nu_s$ as some models show high accuracy but show incorrect relationships between the inputs and output because the machine learning model only built based on the data and do not consider the physical behavior of the model, Therefore, in the proposed GPR model was considered the physical behavior to prove the reliability. In addition, the proposed GPR model was compared with most published models to predict the $\nu_s$ and it has the highest accuracy. After the proposed GPR model was developed to accurately predict the $\nu_s$, the fracture pressure was determined based on the proposed GPR $\nu_s$ model with the previous $\nu_s$ models to evaluate their accuracy on the fracture pressure determinations.

## 2 Methodology

Initially, data was gathered from diverse regions to involve a wide range for developing prediction models for $\nu_s$. The gathered data goes through a cleaning process applying the BAW method to remove outliers. Next, the datasets are utilized with a regression learner to obtain the best algorithm for obtaining $\nu_s$. Subsequently, the cleaned data was divided into three subsets: training, validation, and testing. The training and validation datasets were employed to construct the GPR approach to predict $\nu_s$. After creating the model, trend analysis (TA) was conducted to evaluate the effectiveness of the proposed GPR model. Optimal hyperparameters for the GPR model were selected based on accurate predictions and appropriate input trends. Statistical error analysis (SEA), group error analyses (GEA) and error histograms were carried out when all inputs exhibited suitable relationships.

After successfully establishing the robust and accurate GPR model for $\nu_s$ prediction, the testing dataset was utilized to assess the performance of both the proposed model and previously published models. TAs were also conducted for the previously published models. Subsequently, a comparison was drawn between the proposed GPR model and the previously published models to evaluate their performance. Finally, the fracture pressure based on the previous and GPR Poisson's ratio models was obtained, Fig 1.

## 2.1 Pre-processing

A comprehensive dataset consisting of 1691 records was amassed from some locations, including Malaysia, the United States, Venezuela, India, and Saudi Arabia. The primary aim was to ensure a robust and accurate prediction of $v_s$ operating the GPR model, encompassing a diverse data range. The datasets were obtained from measurements taken during well logging, specifically bulk formation density (RHOB), shear time (DTs), compressional time (DTc), and $v_s$ measurements. The data were collected from different sources in the literature [28]–[33]. The assessment of data quality was discussed clearly in our previous study [18].

A statistical overview of the data, which serves as the foundation for the proposed $v_s$ models is shown in Alakbari et al. [18]. Moreover, the data histograms depicting the distribution of the variables are displayed in Alakbari et al. [18].

Following the data collection process, all input and output parameters underwent a cleaning procedure utilizing the box and whisker (BAW) method for identifying and removing outliers, resulting in data [18].

Histograms depicting the parameters for the clean data used in the $v_s$ model are shown in Alakbari et al. [18]. The relationships between the parameters are depicted in Alakbari et al. [18]. The inputs and corresponding models used to estimate $v_s$ are presented in Alakbari et al. [18]. After the collected data has been cleaned, all parameters are normalized between -1 and 1 [34].

The clean data for the $v_s$ models undergoes a splitting process after normalization. Specifically, 60% of the data is allocated for training the GPR model, while 20% and 20% are assigned

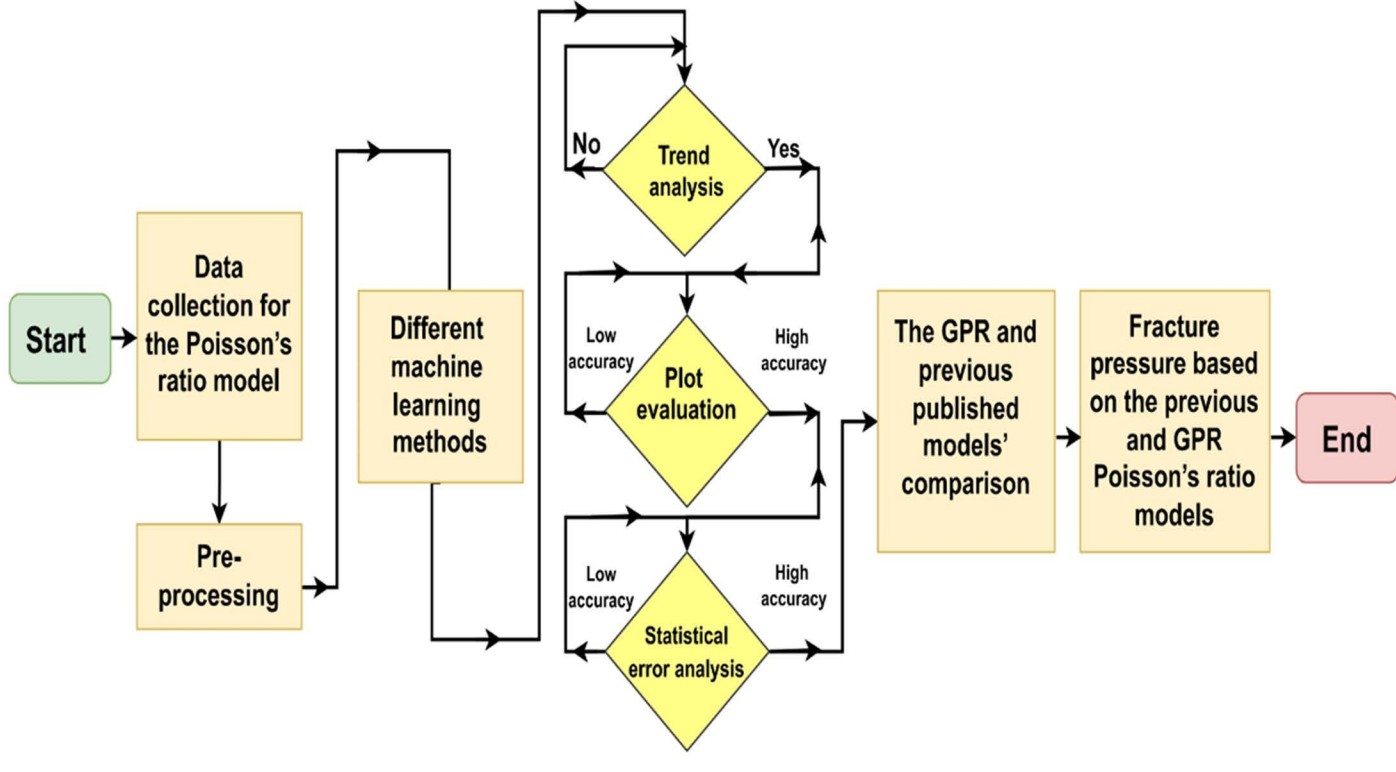

**Fig 1. Methodology flowchart.**

for validation and testing purposes, respectively. To ensure randomness and prevent overfitting and generalization issues, the shuffle method is employed, which involves altering the original order of the data points within each dataset [34].

## 2.2 The Gaussian Process Regression (GPR) model

Regression techniques are widely employed in various engineering applications. polynomial regression and back-propagation neural networks have shown promise in determining parameters and training models. However, these methods are susceptible to overfitting issues [35,36]. The disadvantages of polynomial regression and back-propagation neural networks pay more attention to reducing the error and ignoring the generalization and predictive ability of the model. Therefore, it will show overfitting, which claims that a model fits the training set very well but cannot work on other datasets [37]. Bayesian regression, on the other hand, can address overfitting by identifying a function distribution with prior probabilities for each possible function [38].

An alternative approach is GPR, which can be used for an unlimited number of functions [37,39]. The Gaussian process (GP) is a collection of random variables, any finite number of which has a joint Gaussian distribution. It is known by a mean function and positive definite covariance function [40,37]. The main concept of it is to neglect the form of the regression equation and directly infer the statistical distribution of the functions [41]. GPR offers several advantages, including the ability to present uncertainty metrics for predicting and solving high-dimensional and nonlinear problems, as well as adaptability and generalization capabilities [42–44]. GPR is established to explain the propagation of uncertainties and enhance the efficiency in high-dimensional analysis [45]. The fundamentals and equations for the GPR algorithm were discussed clearly in [46].

Table 1 shows the proposed GPR approach MATLAB code's specifications. The essential parameters are basis and kernel functions which are pure quadratic and squared exponential, respectively. The quasi-newton is applied as the optimizer for parameter estimation. The standard deviation of regularization is 1. In this study, the TA was applied to study the relations

**Table 1. The optimized parameters of the proposed GPR method.**

| Parameter | Description or value |
| --- | --- |
| Basis Function: explicit base on the GPR model | Pure quadratic |
| Kernel Function: a form of the covariance function | Squared exponential |
| Kernel Parameters: initial values for vectors kernel parameters | [3,3 |
| Sigma: initial value for the standard noise deviation of the GPR | 0.020 |
| Standardize: flag to standardize data | 1 |
| Regularization: standard deviation of regularization | 0.5 |
| Computation Method: method to calculate the probability and the gradient of the record | QR (the method based on QR factorization) |
| Distance Method: method to calculate distances between points | fast |
| Active Set Method: active set selection method | random |
| Active set size | 1015 |
| Num Active Set Repeats: number of repetitions | 3 |
| Predict Method: the method used to make predictions | exact |
| Optimizer: optimizer to be used for parameter estimation | quasi newton |

between the parameters to show the effects between the parameters to present the physical behavior (PB) [47–49].

The SEA can be applied to present the approach's performance. Some SEAs are used namely, average percent relative error (APRE), average-absolute percentage relative error (AAPRE), minimum absolute percent relative error (Emin), maximum absolute percent relative error (Emax), root mean square error (RMSE), and standard-deviation (SD), and R. The SEA calculations can be found by using equations 3–8. The GEA was applied in this study to prove the approach's accuracy at different ranges. The functions of SEA are presented in the following:

**Relative deviation error ( $E_i$ ).** The relative deviation error ( $E_i$ ) can be determined as follows:

$$E_i = \frac{Predicted\,E_s - measured\,E_s}{measured\,E_s} \tag{3}$$

i = 1, 2, 3,..., n.

**Average Percentage Relative Error (APRE).** APRE is obtained from equation (4):

$$E_r = \left(\frac{1}{n}\right)\sum_{i=1}^{n} E_i \star 100 \tag{4}$$

**Average Absolute Percentage Relative Error (AAPRE).** AAPRE is given in the following:

$$E_a = \left(\frac{1}{n}\right)\sum_{i=1}^{n} \left|E_i\right| \star 100 \tag{5}$$

**The correlation coefficient (R).** R is estimated from:

$$R = \sqrt{1 - \frac{\sum_{i=1}^{n}[measured\,E_s - predicted\,E_s]}{\sum_{i=1}^{n}[measured\,E_s - \underline{\Delta}E_s]}}$$

Where:

$$\underline{\Delta}E_s = \frac{1}{n}\sum_{i=1}^{N}[measured\,E_s]_i \tag{6}$$

**Standard deviation (SD).** The SD is represented in the following equation:

$$SD = \sqrt{\sum_{i=1}^{n} \frac{[x_{errors} - x_{i\,errors}]^2}{n-1}} \tag{7}$$

**Root Mean Square Error (RMSE).** The RMSE is calculated from equation (6):

$$RMSE = \sqrt{\left[\frac{1}{n}\sum_{i=1}^{n} E_i^2\right]} \tag{8}$$

## 2.3 Application of Poisson's ratio

The Poisson's ratio is used to obtain the fracture pressure by applying the following equations:

$$FP = FG \times D \tag{9}$$

$$FG = PP + [(OBG - PP)(\upsilon/1 - \upsilon)] \tag{10}$$

Where:

FP: fracture pressure, psi.

D: depth, ft.

FG: fracture gradient, psi/ft.

PP: pore-pressure gradient, psi/ft.

OBG: overburden gradient, psi/ft.

ν: Poisson's ratio, dimensionless [50,51].

In this study, the FP was calculated based on the constant values of D = 20000 ft, PP = 0.435 psi/ft, and OBG = 0.845 psi/ft and 338 values of static ν that were determined by using the current proposed (GPR), Khandelwal et al. [13], Ranjbar-Karami et al. [14], Brandås et al. [15], Christaras et al. [52], Feng et al.'s [16], Kumar et al. [12], Gowida et al. [17] models. The same dataset that the testing dataset was used to determine the $ν_s$ for all models. After the FP was obtained based on the different values of $ν_s$ using different methods, the residual error was estimated for all methods. The residual error is the difference between the actual and predicted values of the FP. The actual value of the FP was determined based on the measured $ν_s$. The predicted values of the FP were found based on the predicted $ν_s$ for the different approaches.

## 3 Results

### 3.1 The Ranking of Common Machine Learning Methods for The Prediction of Static Poisson's Ratio ($ν_s$)

The prediction of $ν_s$ is generally achieved using machine learning methods, such as support vector machine (SVM), Gaussian process regression, and ensemble-bagged trees. These methods are applied to a wide range of data collected from different locations and models are developed based on specific data ranges.

The machine learning methods for predicting $ν_s$ were evaluated and ranked based on their low RMSE and high coefficient of determination ($R^2$) values as listed in Table 2, which shows that the top-ranking model for $ν_s$ determination is Gaussian process regression (Exponential GPR), with RMSE, $R^2$, mean squared error (MSE), and mean absolute error (MAE) values of 0.011, 0.95, 0.00012, and 0.0060, respectively. The second-ranked model is Rational Quadratic GPR, with similar performance measures of RMSE, $R^2$, MSE, and MAE. The lowest ranking model is ensemble boosted trees, with RMSE, $R^2$, MSE, and MAE values of 0.021, 0.84, 0.00042, and 0.0176, respectively. Therefore, Exponential GPR is selected as the best model for further enhancement of accuracy. Nonetheless, it is worthwhile to note that the next two models which are also the GPR methods in the table are also likely to behave very closely.

### 3.2 Assessing The GPR approach

Since the GPR is identified above as the optimal machine learning approach for determining $ν_s$. It was selected to conduct further analyses for possible improvements, such as TA, cross-plotting, SEA, and GEA. TA was utilized to identify appropriate relationships between the input parameters and the output. Cross-plotting, error histograms, GEA, and SEA (including measures like R) were employed to assess the accuracy of the proposed GPR models. These analyses collectively provided a comprehensive assessment of the model's robustness and accuracy in predicting $ν_s$.

#### 3.2.1 Trend analysis.

*3.2.1.1 Bulk formation density trend analysis:* Fig 2 shows the $ν_s$ - RHOB's TA for the proposed (optimized) GPR model in this research and several previously published models, along with the experimentally measured values. It can be observed that most of the models including the proposed model give consistent results over a significant range of RHOB showing their

**Table 2. The most common machine learning methods to predict $\nu_s$.**

| No. | Model | RMSE | R² | MSE | MAE | MAPE |
|---|---|---|---|---|---|---|
| 1 | Exponential GPR | 0.011 | 0.95 | 0.00012 | 0.0060 | 0.60 |
| 2 | Rational Quadratic GPR | 0.011 | 0.95 | 0.00012 | 0.0063 | 0.63 |
| 3 | Matern 5/2 GPR | 0.012 | 0.95 | 0.00136 | 0.0068 | 0.68 |
| 4 | Fine_Gaussian_SVM | 0.012 | 0.95 | 0.00014 | 0.0074 | 0.74 |
| 5 | Squared_exponential_GPR | 0.012 | 0.94 | 0.00015 | 0.0072 | 0.72 |
| 6 | Fine_tree | 0.014 | 0.93 | 0.00019 | 0.0082 | 0.82 |
| 7 | Ensemble_bagged_trees | 0.014 | 0.93 | 0.00019 | 0.0086 | 0.86 |
| 8 | Quadratic_SVM | 0.014 | 0.93 | 0.00019 | 0.0090 | 0.90 |
| 9 | Cubic_SVM | 0.014 | 0.93 | 0.00019 | 0.0086 | 0.86 |
| 10 | Medium_tree | 0.014 | 0.93 | 0.00020 | 0.0089 | 0.89 |
| 11 | Medium_Gaussian_SVM | 0.014 | 0.93 | 0.00020 | 0.0087 | 0.87 |
| 12 | Coarse_tree | 0.016 | 0.91 | 0.00025 | 0.0106 | 1.06 |
| 13 | Interactions_linear_regression | 0.018 | 0.87 | 0.00034 | 0.0127 | 1.27 |
| 14 | Stepwise_linear_regression | 0.018 | 0.87 | 0.00034 | 0.0127 | 1.27 |
| 15 | Coarse_Gaussian_SVM | 0.019 | 0.86 | 0.00036 | 0.0122 | 1.22 |
| 16 | Linear_regression | 0.020 | 0.85 | 0.00040 | 0.0147 | 1.47 |
| 17 | Robust_linear_regression | 0.020 | 0.85 | 0.00041 | 0.0146 | 1.46 |
| 18 | Linear_SVM | 0.020 | 0.84 | 0.00041 | 0.0146 | 1.46 |
| 19 | Ensemble_boosted_trees | 0.021 | 0.84 | 0.00042 | 0.0176 | 1.76 |

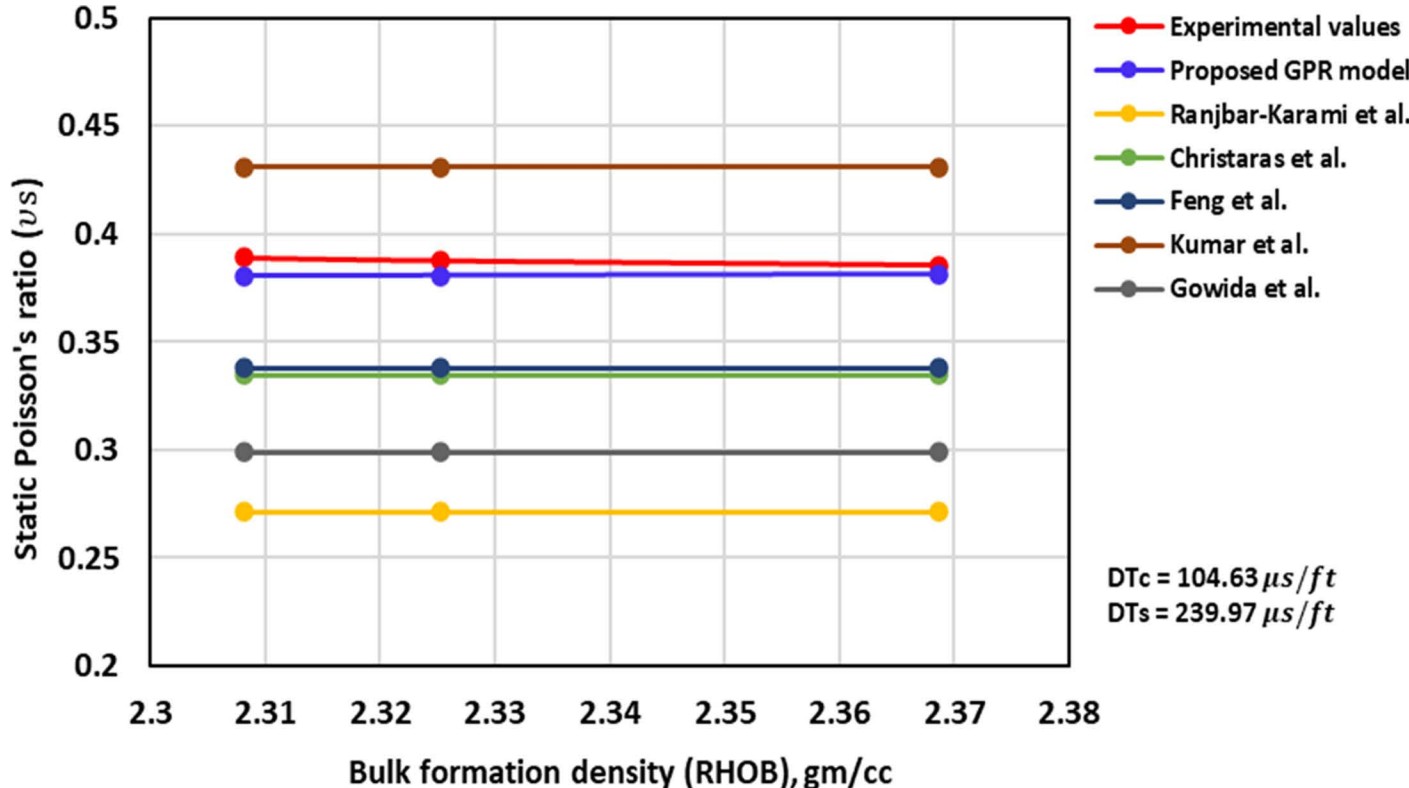

**Fig 2. The comparison of $\nu_s$ – RHOB trends models with experimental measurements.**

robustness. However, the proposed model stands out from the rest in that it's $\nu_s$ values are very close to the experimental ones, assuring its high accuracy compared to other models. The experimental data shows a decline in $\nu_s$ values with increasing RHOB. The optimized GPR model also captures this trend, i.e., the $\nu_s$ decreasing as the RHOB increases, albeit to a much smaller extent not easily discernable in Fig 2. The similarity in trends indicates that it follows the proper physical behaviour (PB). The other models, on the other hand, stayed constant contrary to the experimental results.

Fig 3 shows the TA of the $\nu_s$ – RHOB within the range of data in this study. The trend behavior is like the expected behavior as observed through experiments. Thus, it can be concluded that the optimized GPR method exhibits the appropriate PB in capturing the relationship between the RHOB and the $\nu_s$.

*3.2.1.2 Compressional time trend analysis:* Figs 4–6 illustrate the TAs of the $\nu_s$ -DTc for the optimized GPR and previously published methods. The TA of the $\nu_s$ -DTc for the models with the measurement values, are illustrated in Figs 4 and 5. The measurement values imply that as DTc increases, $\nu_s$ decreases. The proposed GPR model also exhibits a similar trend, with the $\nu_s$ decreasing as the DTc increases, thereby confirming the correct relations (Figs 4 and 5). Additionally, the $\nu_s$ values predicted by the proposed GPR model closely align with the experimental measurements. However, the methods proposed by others such as Christaras et al. [52] also follow the proper relations, but their $\nu_s$ values deviate from the experimental measurements (Figs 4 and 5). In contrast, the model proposed by Khandelwal et al. [13] suggests that the $\nu_s$ increase with an increase in the DTc in Alakbari et al. [18]. Fig 6 shows the TA of the DTc at its range of the optimized GPR approach. The optimized GPR approach captures the

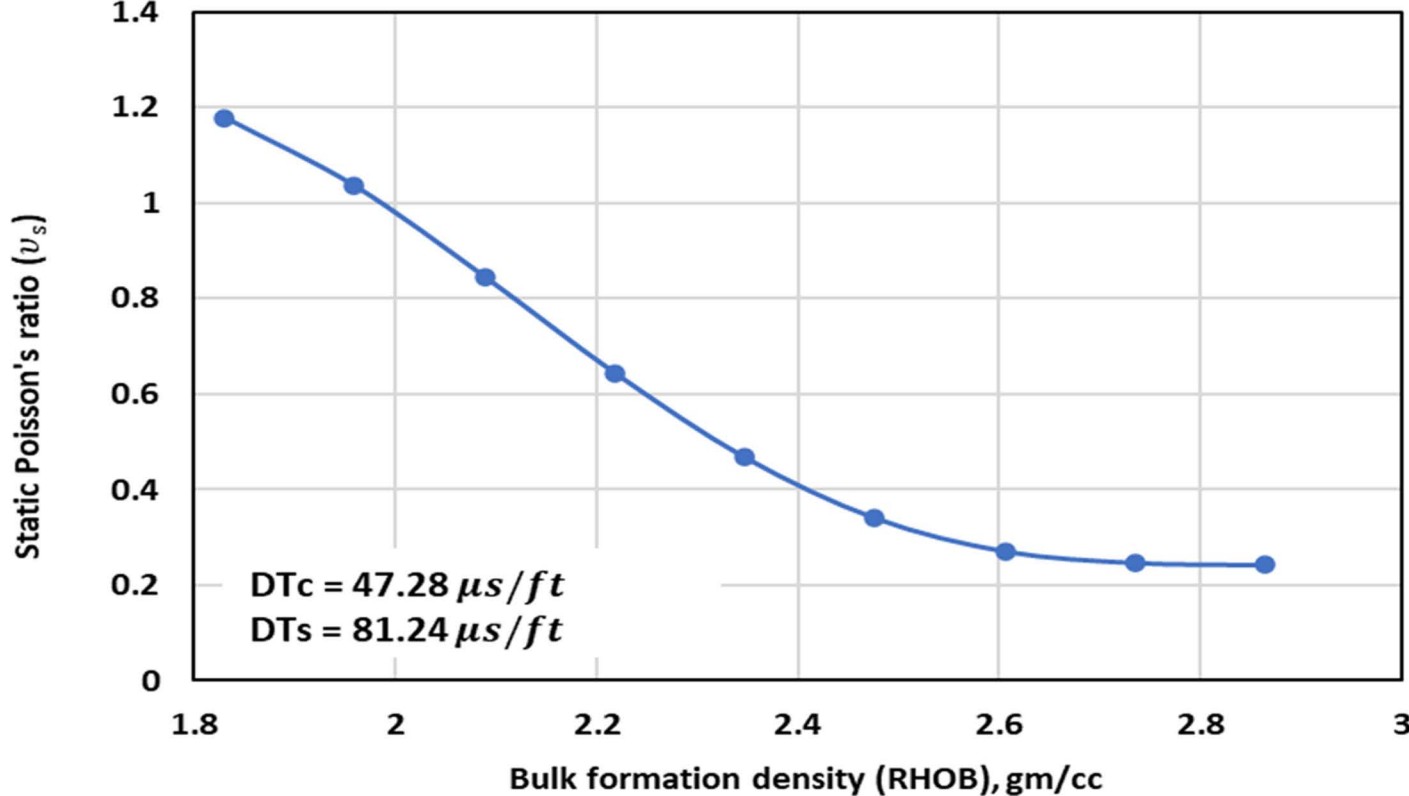

**Fig 3. Trend analysis of the $\nu_s$ – RHOB in the range of data found in the dataset used for the optimized GPR.**

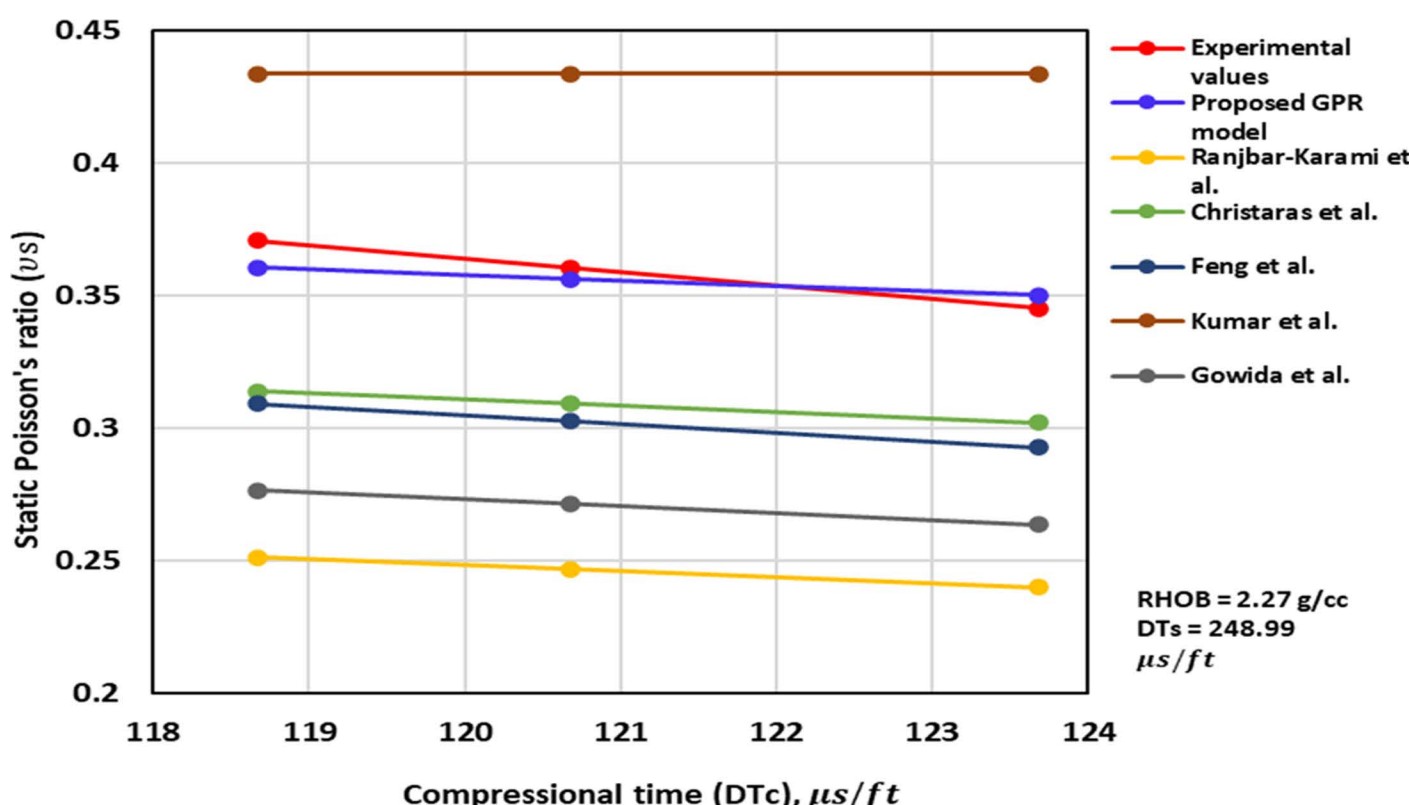

**Fig 4.** The comparison of $v_s$ – DTc trends models with experimental measured values for RHOB = 2.27 g/cc and DTs = 248.99 us/ft.

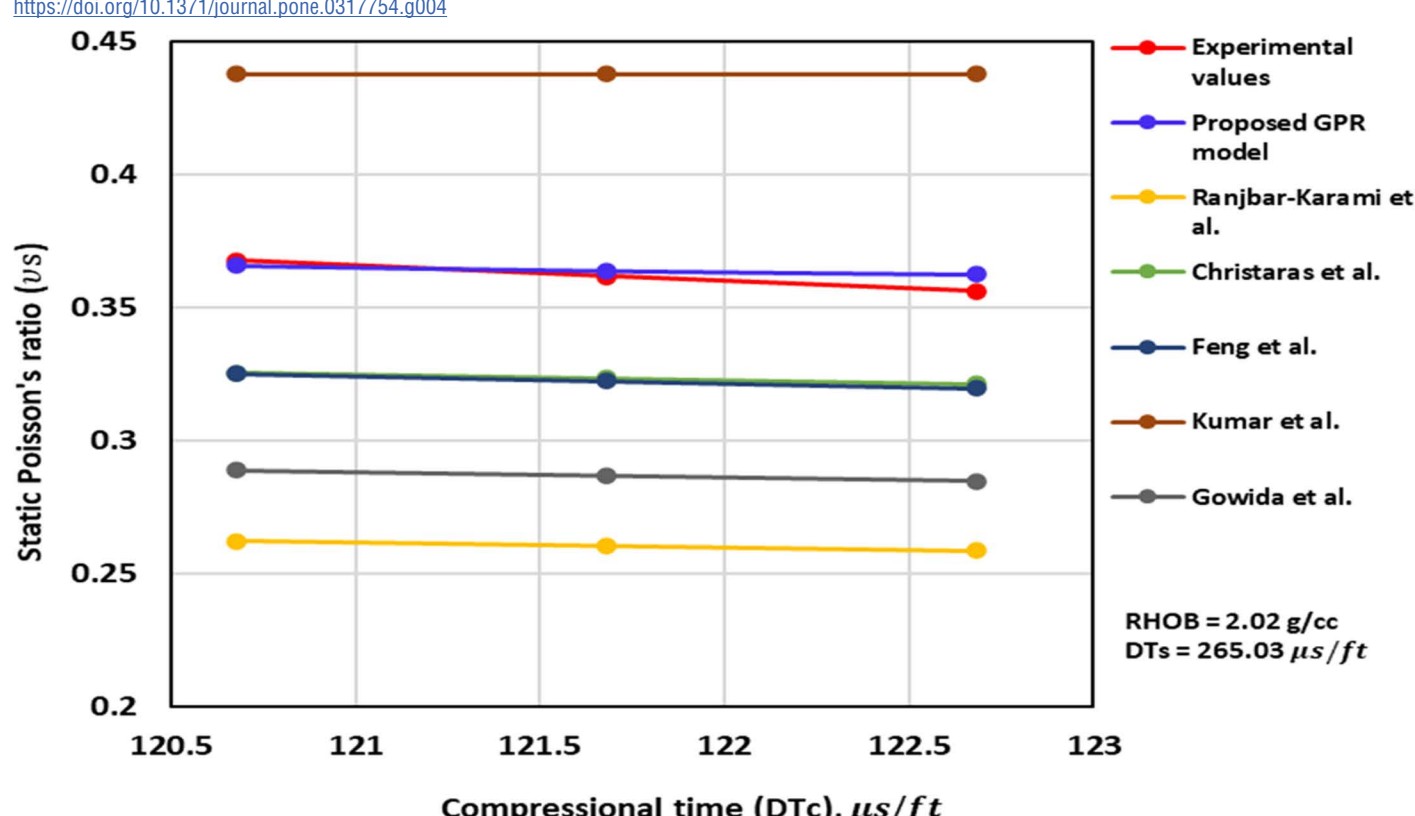

**Fig 5.** The comparison of $v_s$ – DTc trends models with experimental measured values for RHOB = 2.02 g/cc and DTs = 265.03 us/ft.

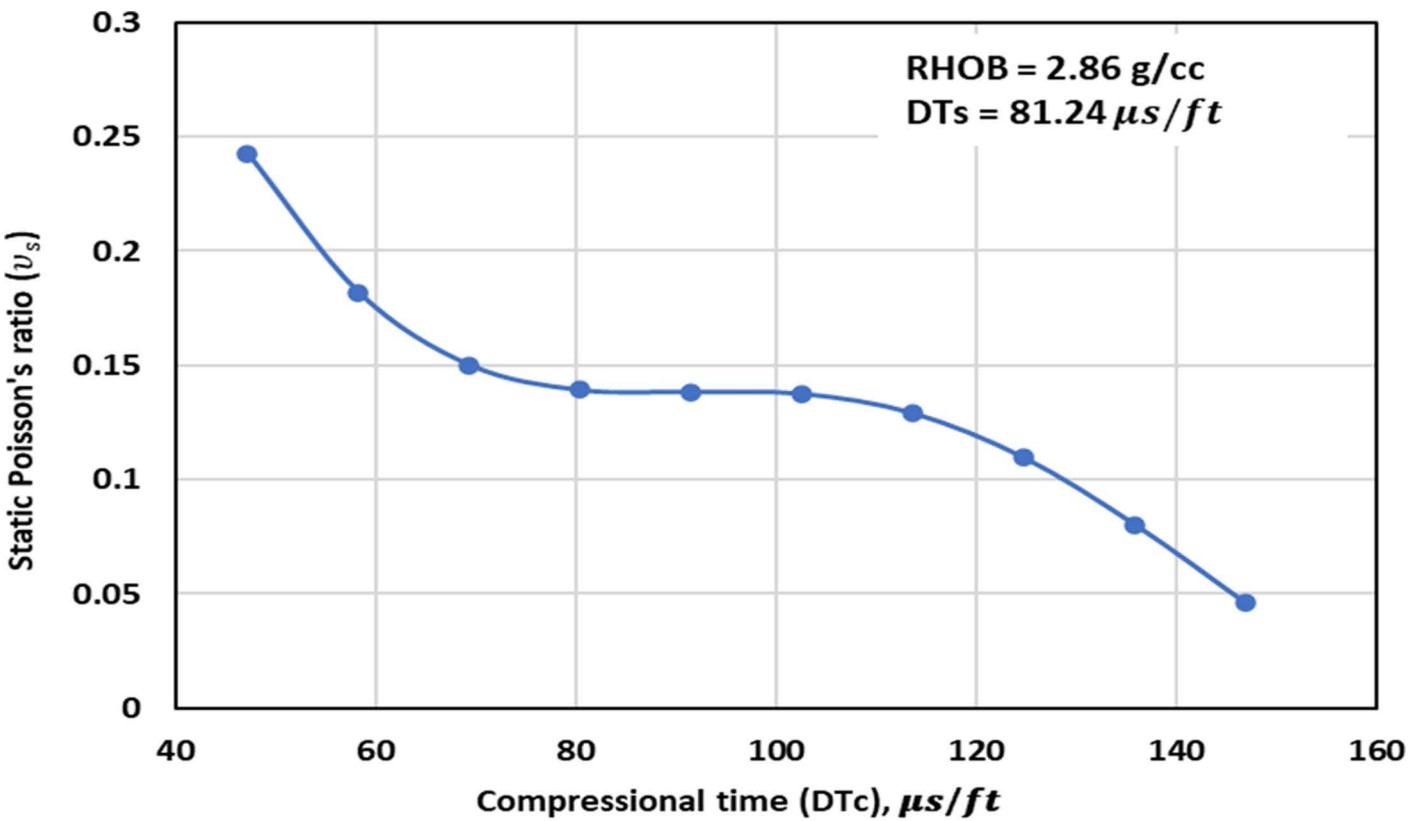

**Fig 6. Trend analysis of the $v_s$ – DTc in the range found in the dataset used for the optimized GPR.**

appropriate trend for the $v_s$-DTc within its range, thus demonstrating the correct PB (Fig 6). Fig 6 displays the optimized GPR approach that has the proper relations to prove the accurate PB.

*3.2.1.3 Shear time trend analysis:* The $v_s$ – DTs trend analysis is depicted in Figs 7 to 9. Figs 7 and 8 illustrate the TA of the $v_s$ – DTs for the previously published and proposed models, along with the measurement values. These Figs demonstrate that an increase in the DTs corresponds to an increase in the $v_s$. The proposed GPR model accurately captures the correct relationship between the DTs and the $v_s$. Similarly, the models proposed by others namely, Christaras et al. [52] exhibit the correct trend of the $v_s$ increasing with an increase in the DTs (Figs 7 and 8). However, Brandås et al. [15] method displays an incorrect trend, suggesting that an increase in the DTs leads to a reduction in the $v_s$ (Alakbari et al. [18]). Khandelwal et al. [13] method indicates that the $v_s$ remain constant as the DTs change Alakbari et al. [18]. The optimized GPR approach displays the accurate trend when the DTs with its range (Fig 9).

Therefore, the proposed GPR model exhibits the appropriate PB. The TAs of all inputs of the optimized GPR approach prove the accurate relations. Consequently, the proposed GPR approach captures the appropriate PB for all parameters.

**3.2.2 Optimized model cross-plotting.** The $v_s$ data predicted from the optimized GPR approach is cross-plotted against the measured values in Figs 10 to 12 for the various datasets: Fig 10 illustrates the training dataset, Fig 11 is for the validation dataset, and Fig 12 is the testing dataset. All three cross-plots show proximity to the red line, indicating the high accuracy of the proposed model. These cross-plotting Figs serve as evidence that the optimized GPR approach achieves excellent performance.

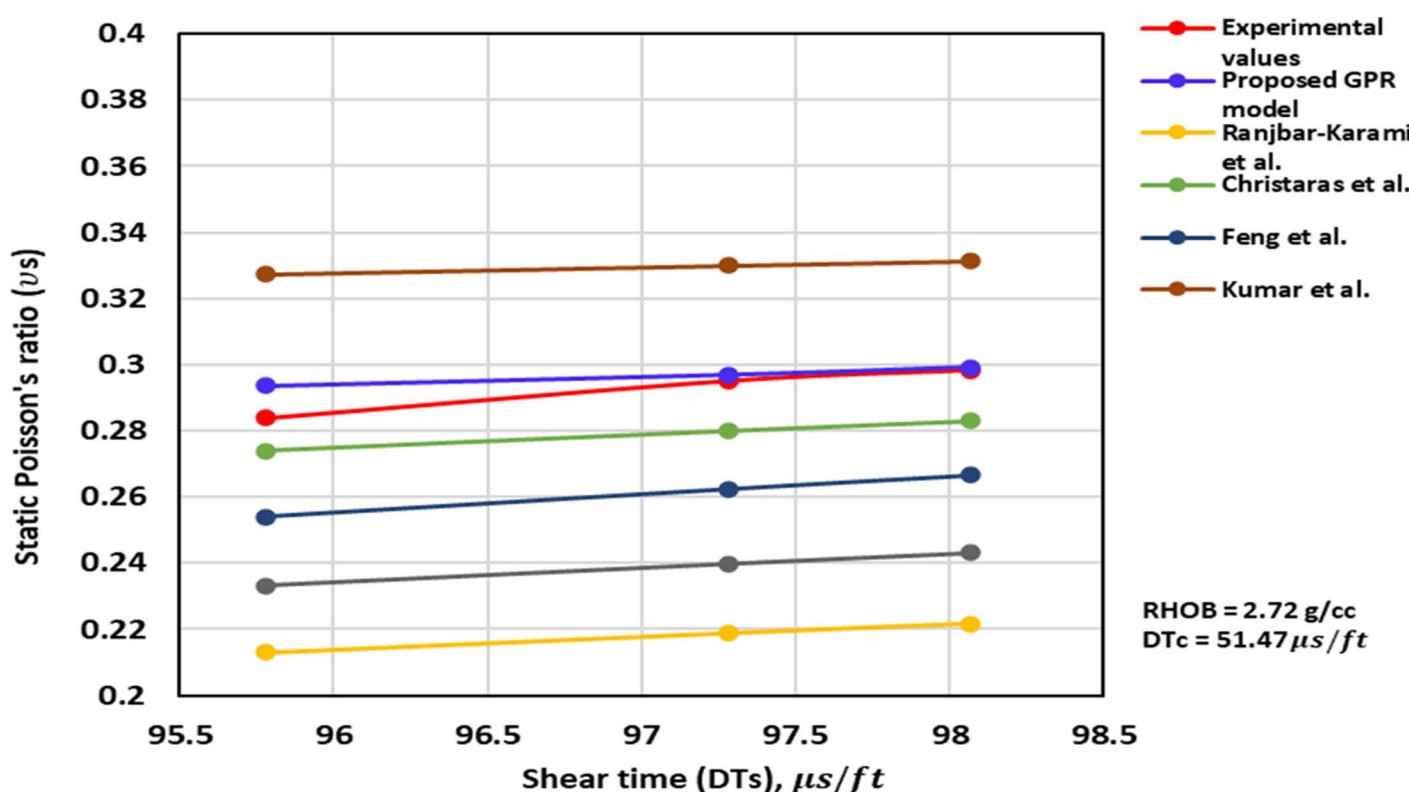

**Fig 7.** The comparison of $v_s$ – DTs trends models with experimental measured values for RHOB = 2.72 g/cc and DTc = 51.47 us/ft.

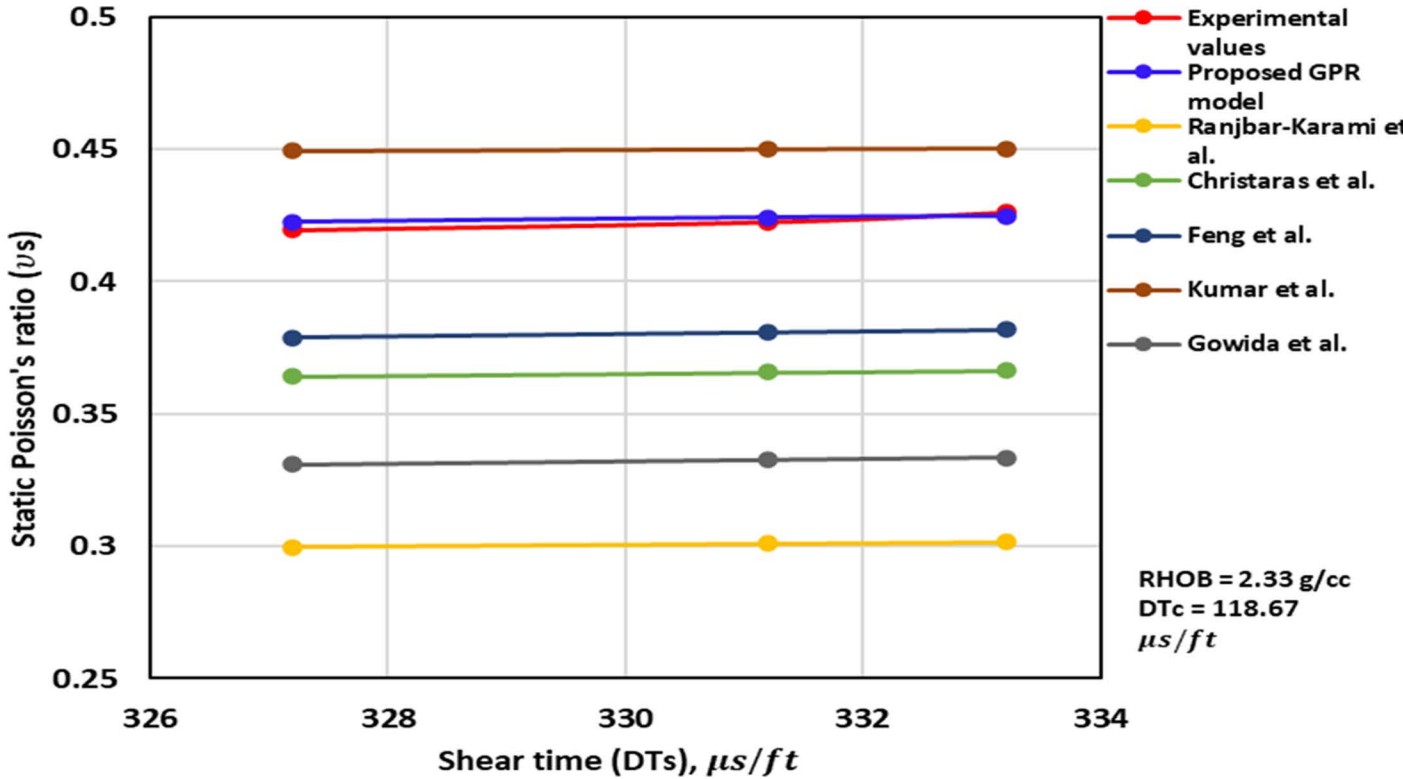

**Fig 8.** The comparison of $v_s$ – DTs trends models with experimental measured values for RHOB = 2.33 g/cc and DTc = 118.67 us/ft.

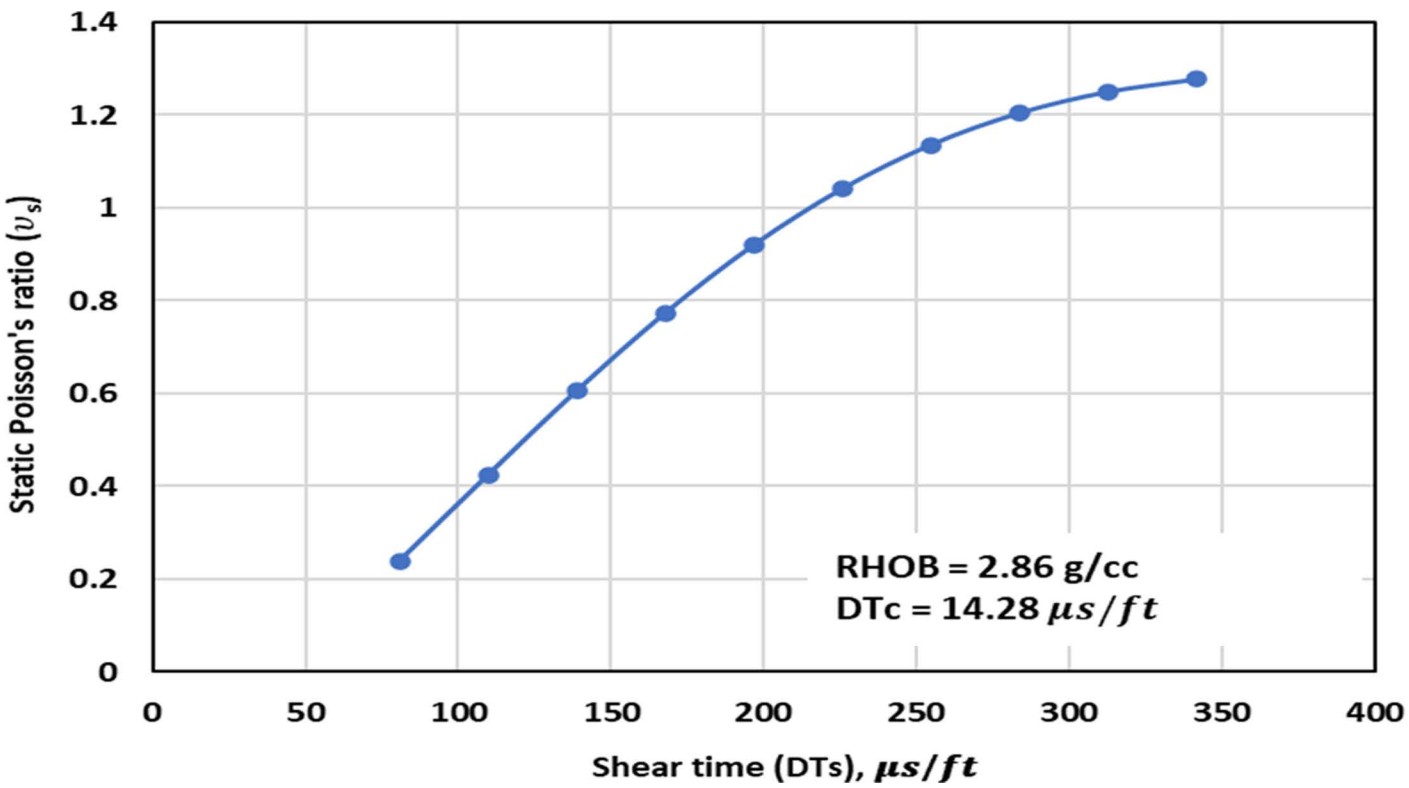

**Fig 9. Trend analysis of the $v_s$ – DTs in the range found in the dataset used for the optimized GPR.**

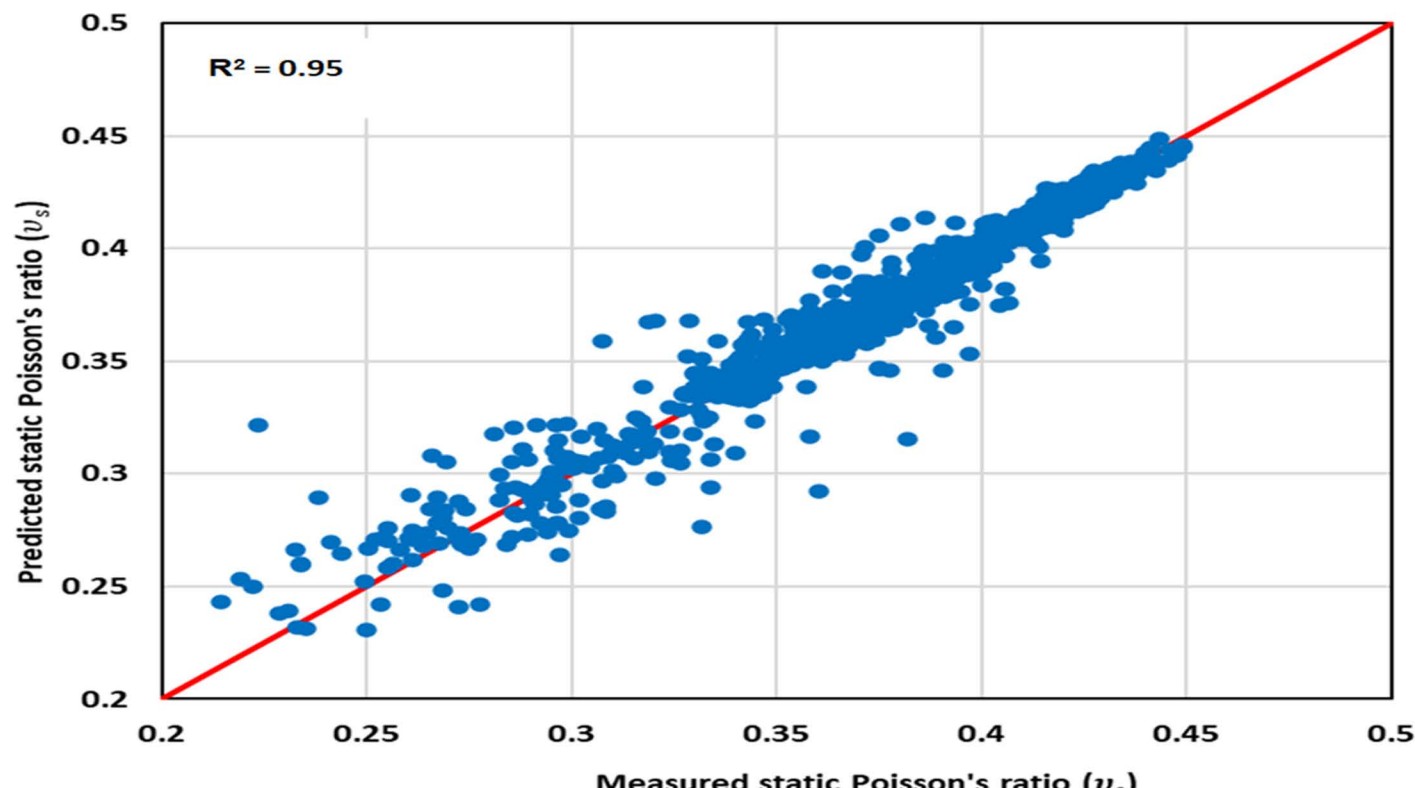

**Fig 10. The optimized GPR approach's cross-plotting for the training dataset.**

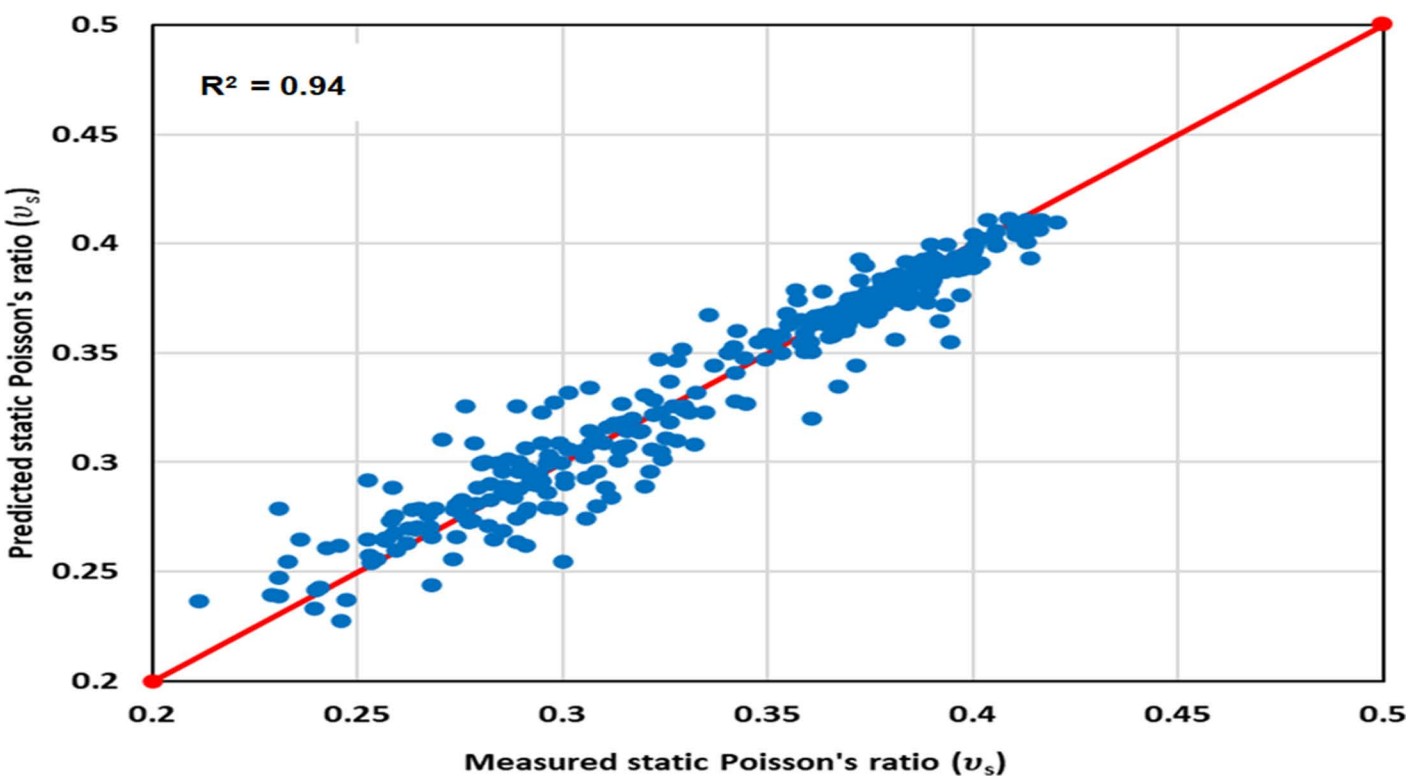

**Fig 11. The optimized GPR approach's cross-plotting for the validation dataset.**

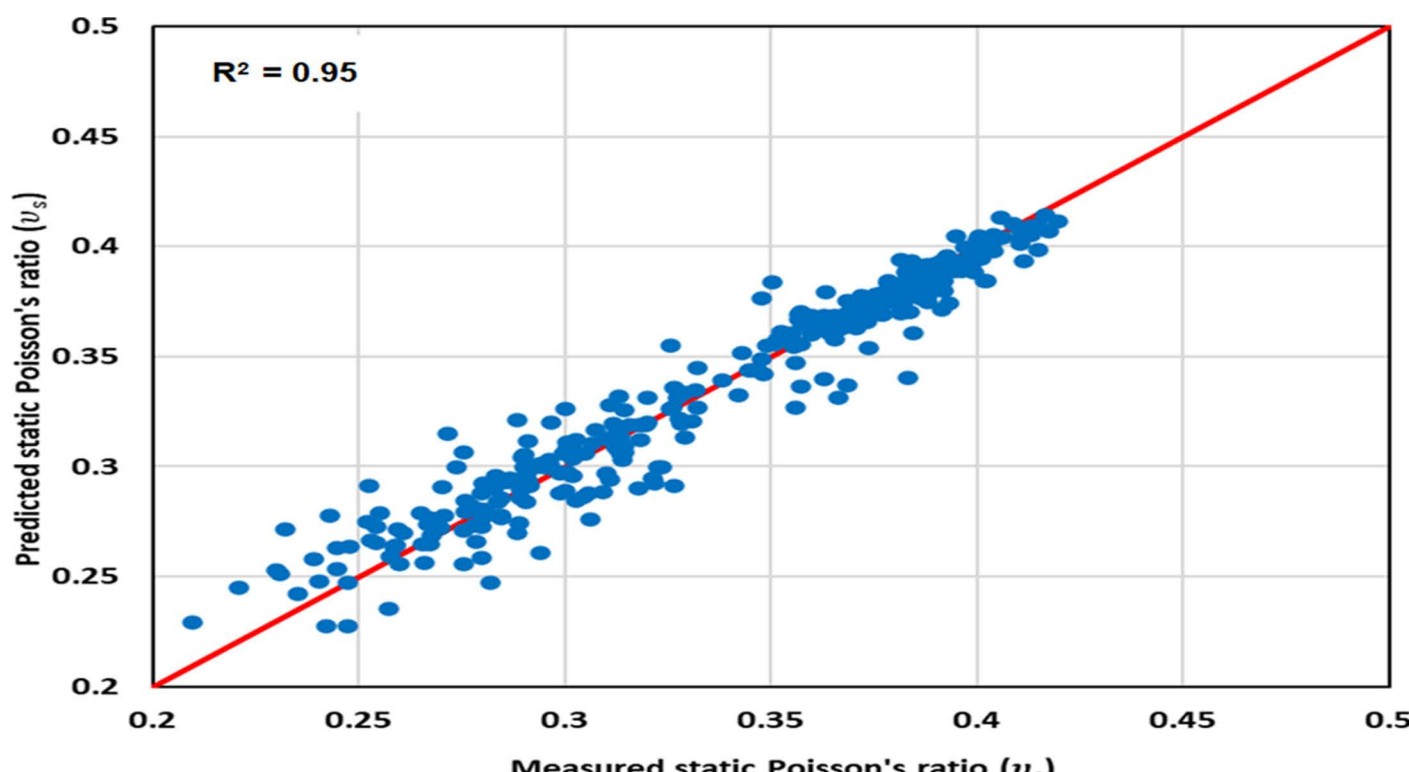

**Fig 12. The optimized GPR approach's cross-plotting for the testing dataset.**

**3.2.3 Optimized model statistical error analyses (SEA).** Statistical error analyses (SEA) were conducted to evaluate the accuracy of the optimized GPR approach in determining $v_s$, including APRE, AAPRE, Emax, Emin, RMSE, R, and STD. The primary indicators employed to evaluate the methods' accuracy were R and AAPRE. Table 3 presents the AAPRE and R values for the training, validation, and testing datasets, which are 2.514%, 3.807%, and 3.773% for AAPRE, and 0.95, 0.94, and 0.95 for $R^2$, respectively. These values demonstrate that the datasets exhibit AAPRE and $R^2$ values that are closely aligned with the actual values, confirming the accurate prediction of $v_s$ by the proposed or optimized GPR model without any over-fitting or under-fitting problems. The RMSE values for the training, validation, and testing datasets are 0.0043, 0.0056, and 0.0058, respectively in Table 3. The SEAs affirm that the optimized GPR approach achieves a satisfactory performance across various datasets. Consequently, the optimized GPR approach exhibits a strong potential for general applicability.

Fig 13 depicts measured and predicted estimates of all three datasets and demonstrates a close correspondence between the measured and predicted $v_s$ for all datasets. This alignment between the measured and predicted $v_s$ confirms the high accuracy of the proposed GPR model in effectively predicting $v_s$.

**3.2.4 Optimized model error histograms.** Figs 14–16 show the histograms depicting the errors (percentage relative error) of the proposed GPR approach of the training, validation, and testing datasets. The histograms illustrate that most data points in all datasets exhibit nearly zero percentage error, as demonstrated in Figs 14, 15 and 16. These error histograms provide strong evidence that the proposed GPR model is capable of accurately predicting $v_s$.

**3.2.5 Group error analysis.** Figs 17–19 present the group error analysis (GEA) of inputs, comparing the optimized GPR approach's accuracy with the previously published methods across various ranges of inputs. The GEA specifically focuses on the RHOB and is depicted in Fig 17, which shows that the optimized GPR approach achieves a superior accuracy with less than 5% AAPRE across all RHOB, whereas most of the other previous models showed much lower accuracy with AAPRE values ranged from 10-30%.

The GEA of DTc is presented in Fig 18 showing that the optimized GPR approach has less than 5% AAPRE for all ranges. However, the previous models display 8-45% AAPRE for most data ranges, namely Feng et al.'s [16] model displays AAPRE values of more than 10% across most ranges, with the model being constructed using DTc values of 54.9-72.55 µs/ft, as shown in Fig 18. Some models such as, Gowida et al. [17] model exhibit AAPRE values of 20-25% at various ranges, as presented in Fig 18.

The DTs GEA is depicted in Fig 19. Fig 19 illustrates that the optimized GPR approach achieves less than 5% AAPRE in the majority ranges. Most published models presented more than 7-40% for the most data ranges, like Gowida et al. [17] models exhibit AAPRE values of 20-25% at various ranges, as presented in Fig 19. They developed their models based on DTs values of 73.19-145.60 and 40-75 µs/ft.

**Table 3. Statistical error analyses of the proposed or optimized GPR model.**

| Dataset | APRE (%) | AAPRE (%) | Emax. (%) | Emin. (%) | RMSE | $R^2$ | STD |
|---|---|---|---|---|---|---|---|
| Training | -0.123 | 2.00 | 43.76 | 0.003 | 0.0043 | 0.95 | 0.011 |
| Validation | -0.043 | 2.87 | 20.80 | 0.022 | 0.0056 | 0.94 | 0.130 |
| Testing | -0.058 | 2.73 | 17.00 | $8.700 \times 10^{-14}$ | 0.0028 | 0.95 | 0.012 |
| whole | 0.094 | 2.32 | 43.76 | 0 | 0.0042 | 0.95 | 0.011 |

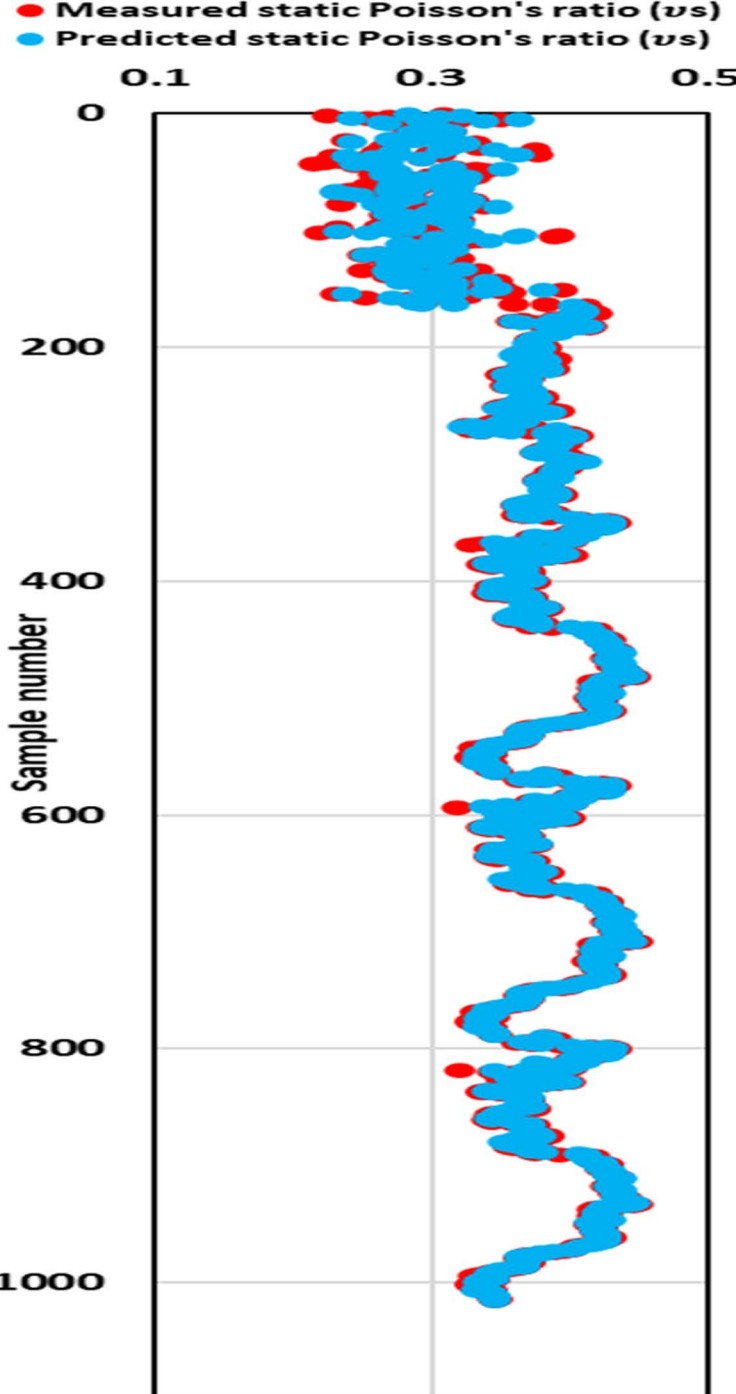

**Fig 13. Comparison of $v_s$ measured and predicted estimates of the optimized GPR approach for (a) training, (b) validation, and (c) testing datasets.**

## 3.3 The optimized model's Comparison with previous models

**3.3.1 Cross-Plotting Comparison.** Fig 20 (a) and (b) show the cross plots of the proposed GPR model along with published models. As shown in Fig 20 (a), most values of the predicted $v_s$ close to the measured values and became in the 45º red line to indicate that the proposed GPR

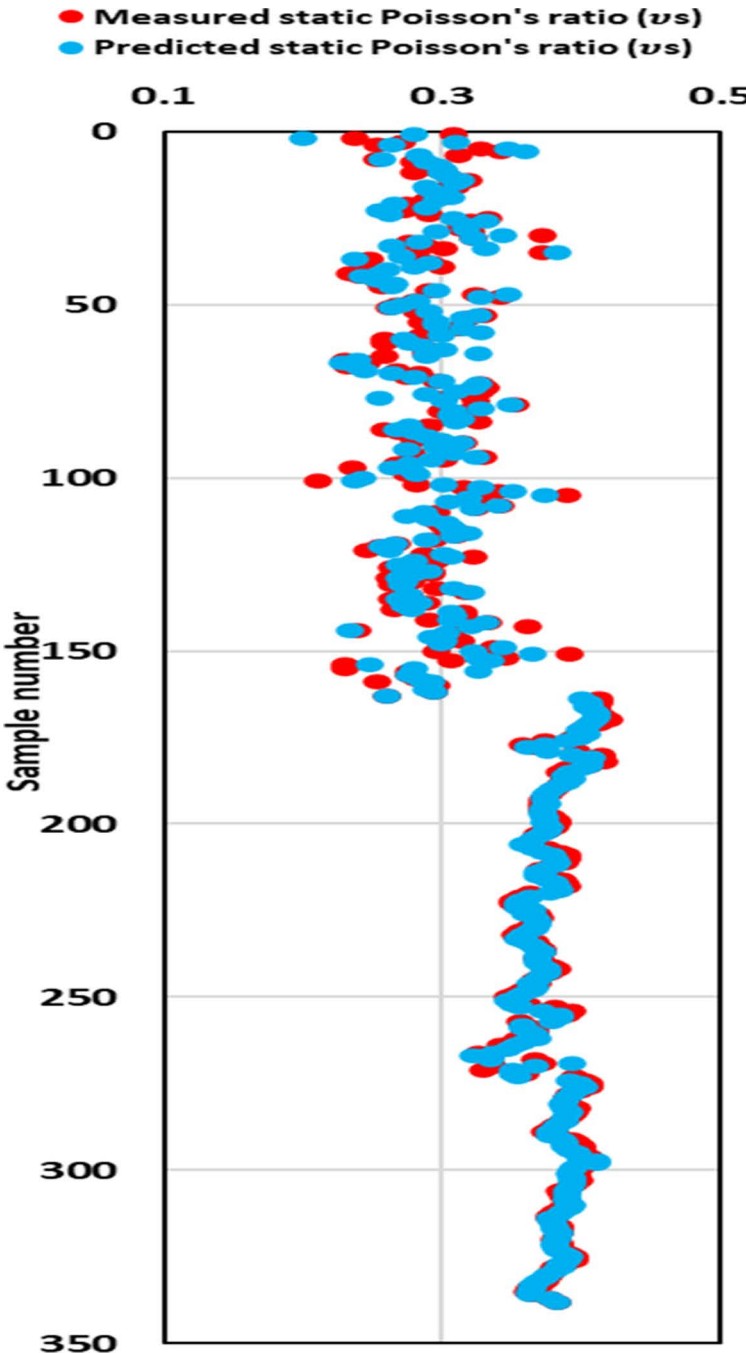

**Fig 13.** Continued.

model has high accuracy compared to the previous models. The second model after the proposed GPR model is Christaras et al. [52] model to find the $\nu_s$. However, Khandelwal et al [13], Brandås et al. [15], and Kumar et al. [12] models have most values of the predicted $\nu_s$ different from the measured or actual $\nu_s$ as their predicted and measured $\nu_s$ are not in the 45-degree red line as shown in Fig 20 (b). Therefore, these models have low accuracy to predict the $\nu_s$.

**3.3.2 Statistical error analyses comparison.** The performance of the optimized GPR and previous approaches was evaluated utilizing SEA and presented in Fig 21 (a, b). The

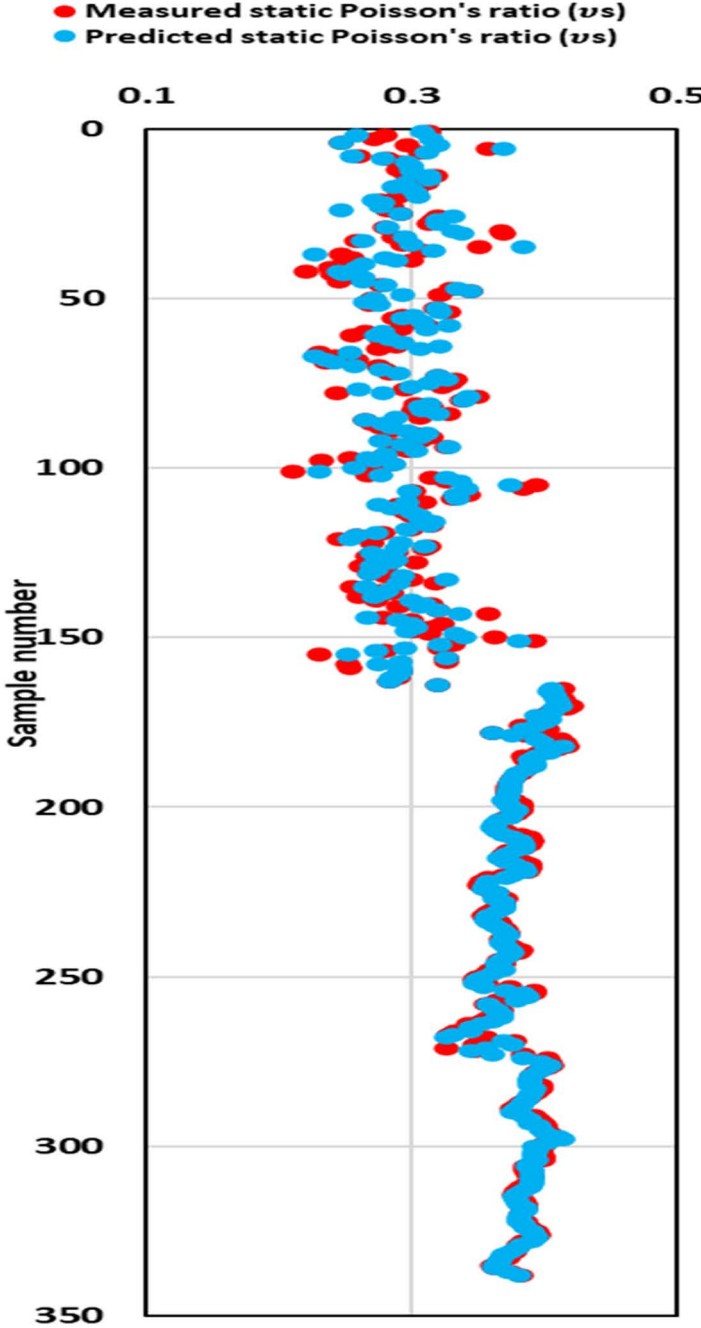

**Fig 13.** Continued.

approaches were listed based on indicators such as $R^2$ and AAPRE. The optimized GPR approach shows the highest accuracy, with 0.95 $R^2$ and 2.73% AAPRE which is the lowest AAPRE. However, the previous models have $R^2$ of (0.45-0.88) and AAPRE of (10.65-127.1) %.

In addition, other SEAs including APRE, Emax, Emin, SD, and RMSE were also calculated for all approaches to compare the performance. Fig 22 (a, b) is a bar graph showing APRE, Emax, Emin, and RMSE values for all the models. The respective values for the optimized model were -0.058%, 17.0%, 8.7e-14%, and 0.0028. However, the previous models have APRE,

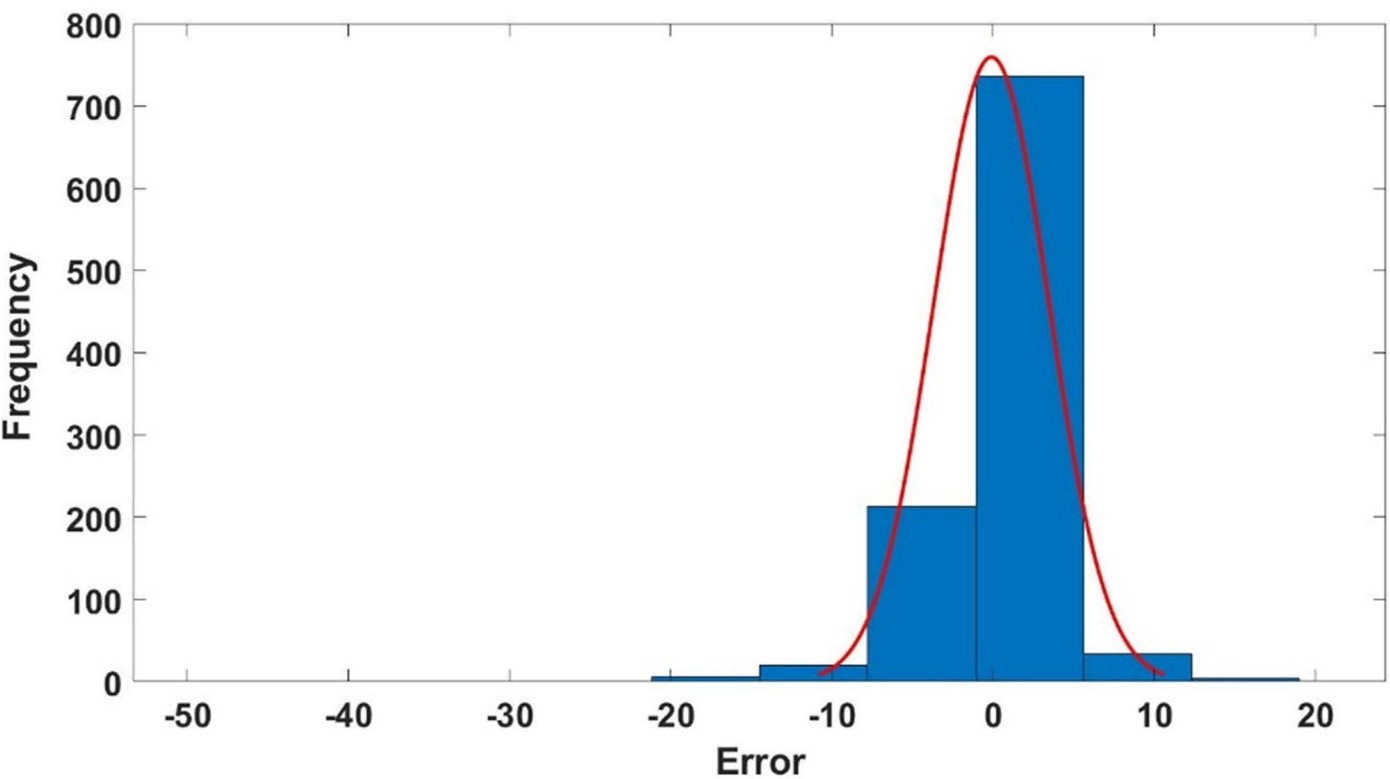

**Fig 14. The optimized GPR approach's error histogram of the training dataset.**

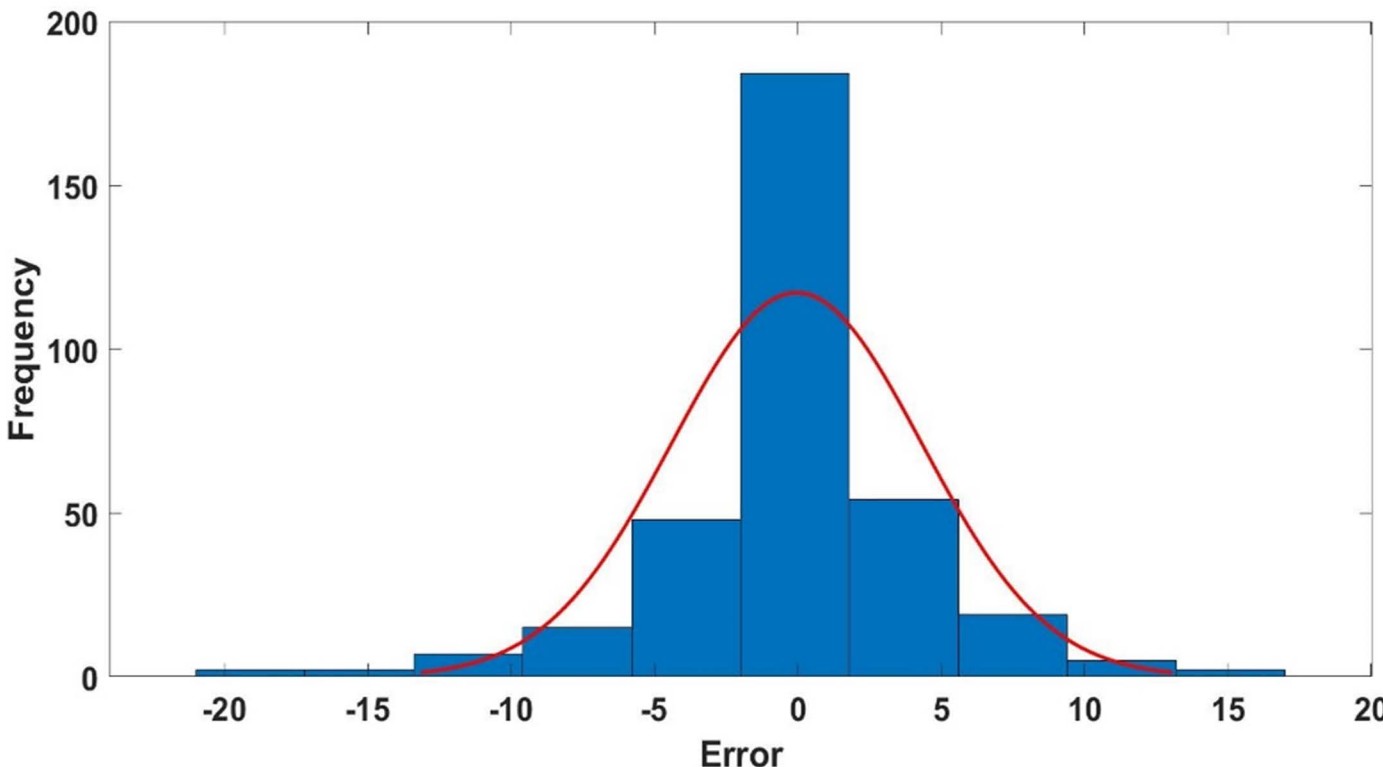

**Fig 15. The optimized GPR approach's error histogram of the validation dataset.**

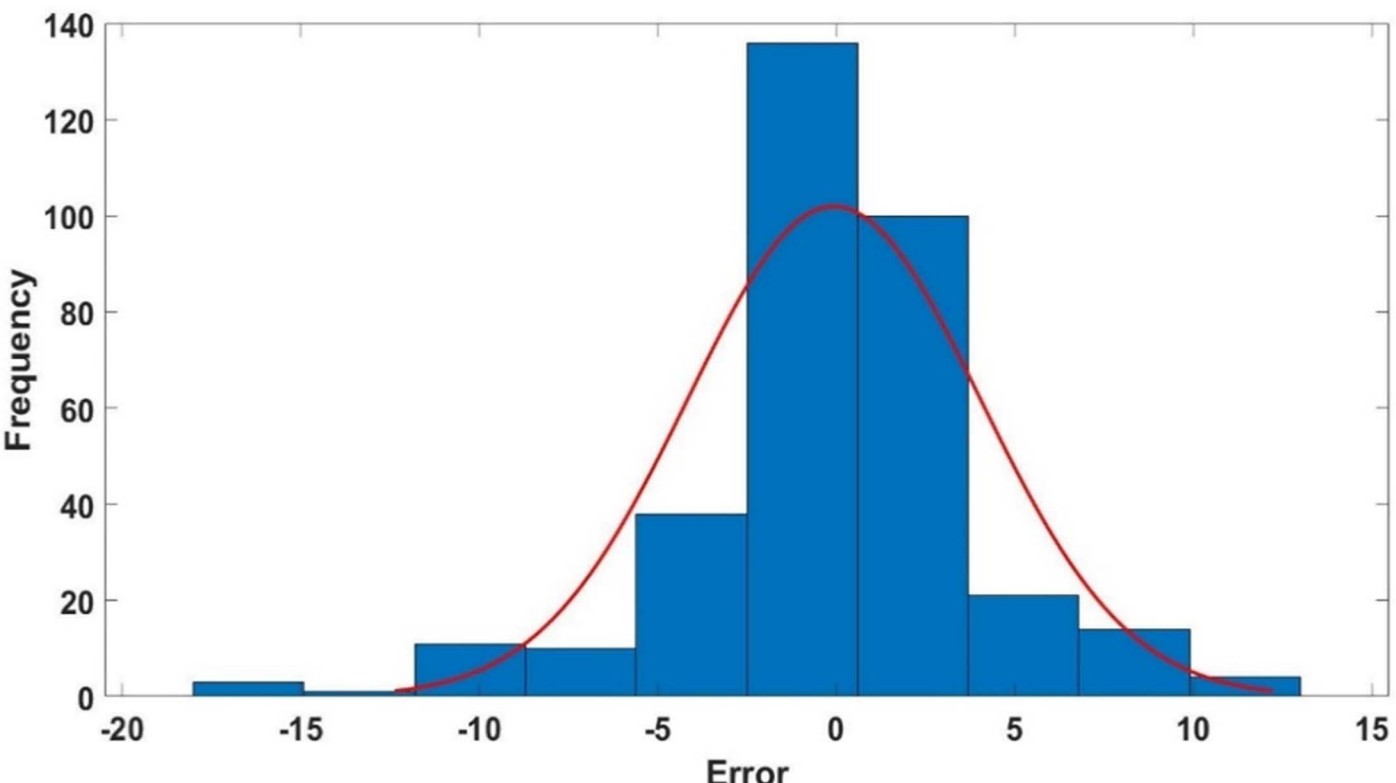

**Fig 16. The optimized GPR approach's errors histogram of the testing dataset.**

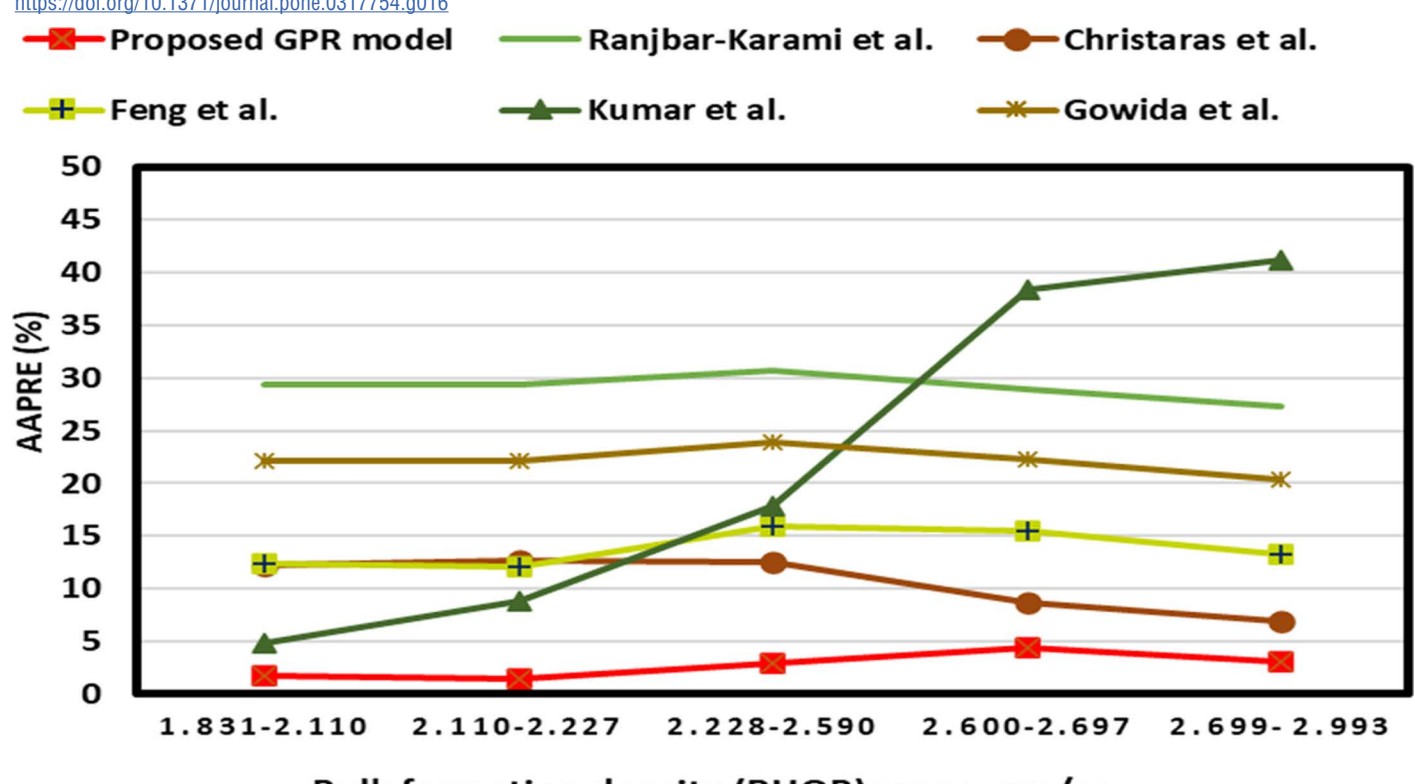

**Fig 17. Group error analysis of bulk formation density for the proposed GPR and some previous approaches.**

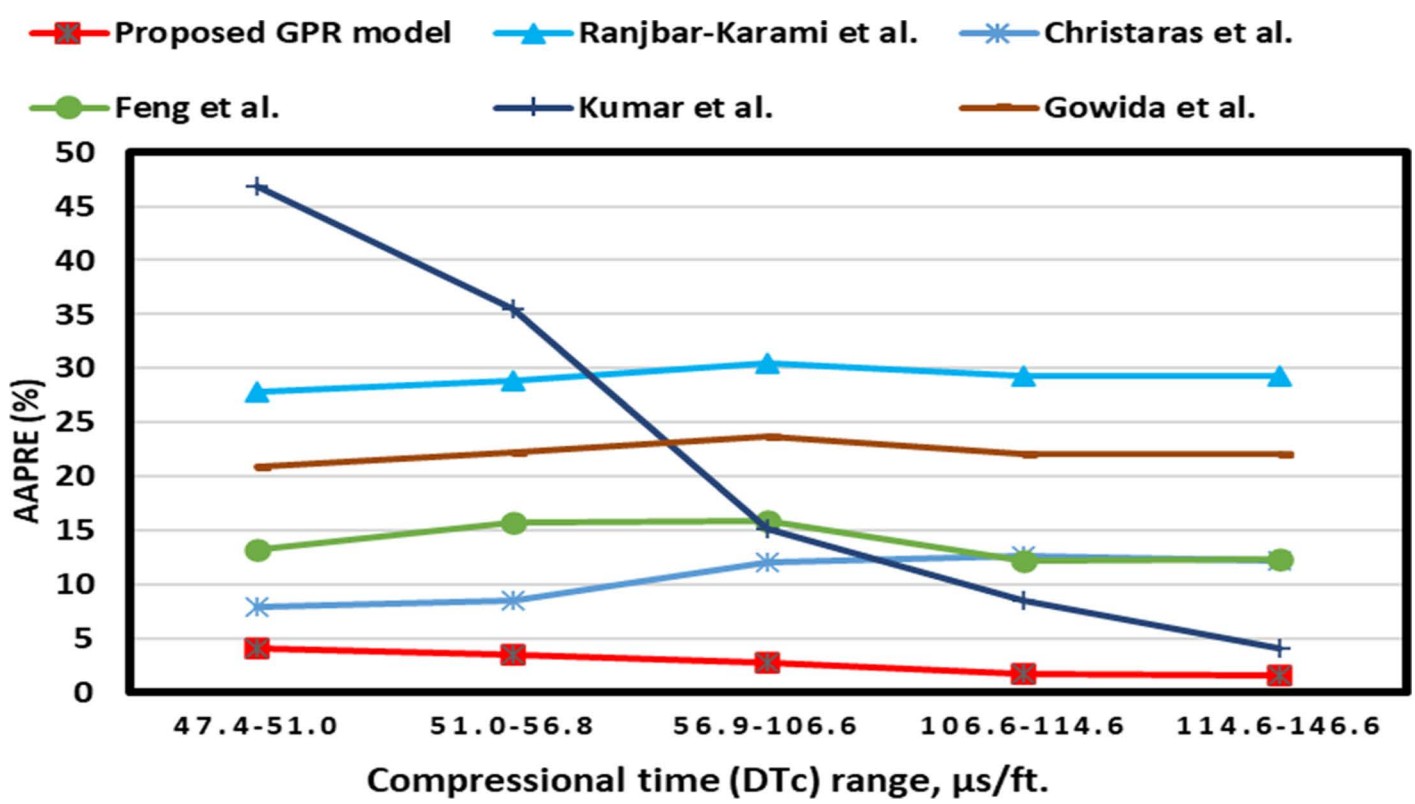

**Fig 18. Group error analysis of compressional time for the proposed GPR and some previous approaches.**

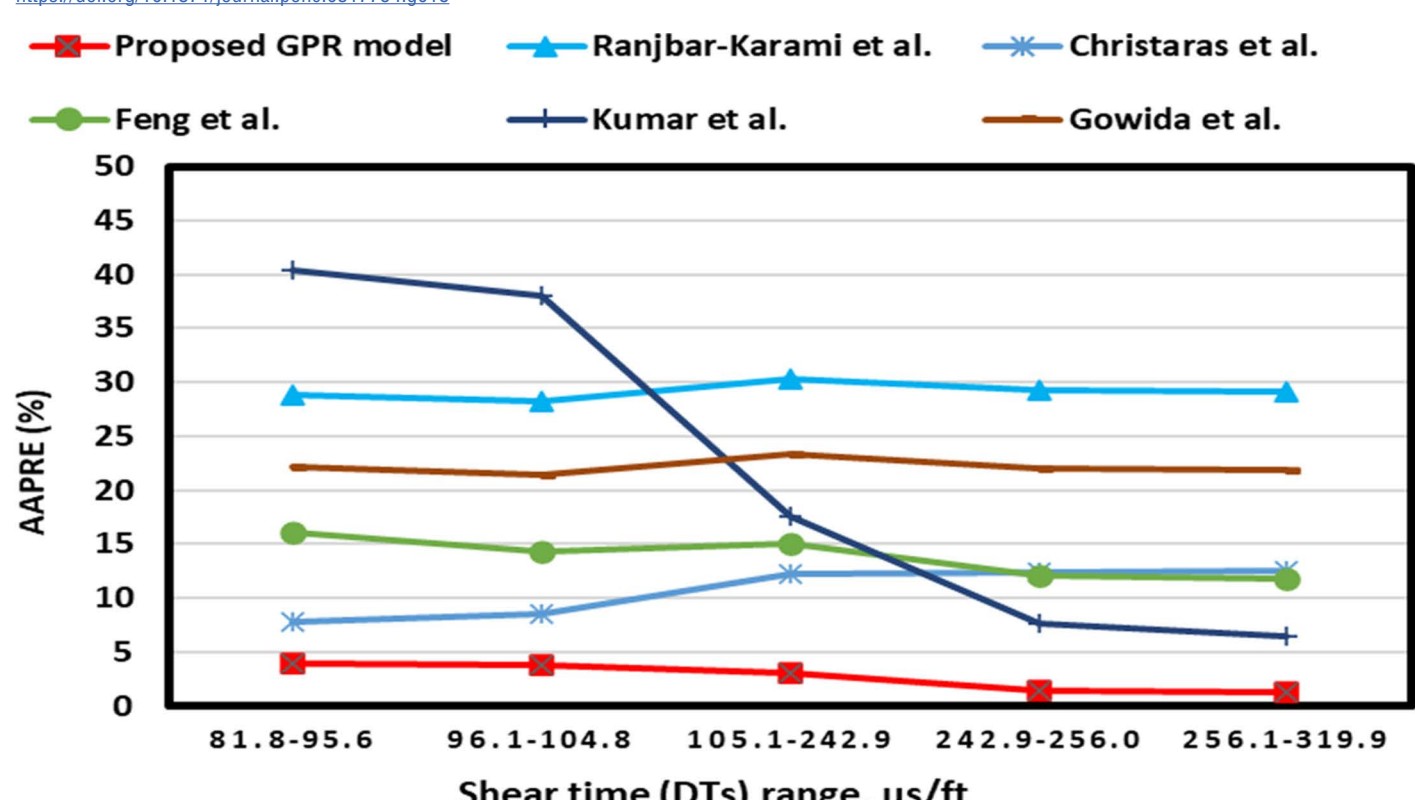

**Fig 19. Group error analysis of shear time for the proposed GPR and some previous approaches.**

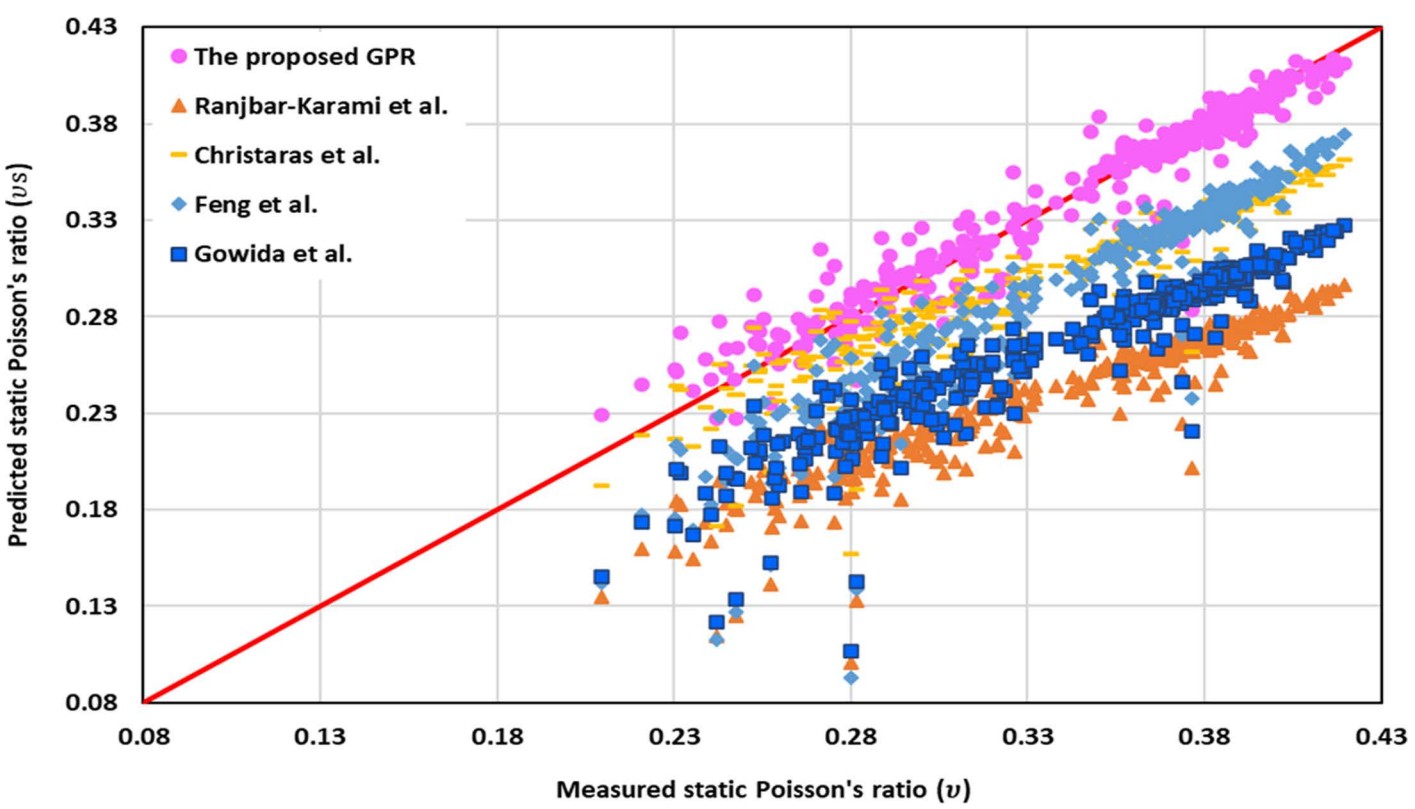

**Fig 20.** Cross-plots of (a) the proposed GPR, Ranjbar-Karaml et al. [14], Christaras et al. [52], Feng et al. [16], Gowida et al. [17] models and (b) Khandelwal et al. [13], Brandås et al. [15], and Kumar et al. [12].

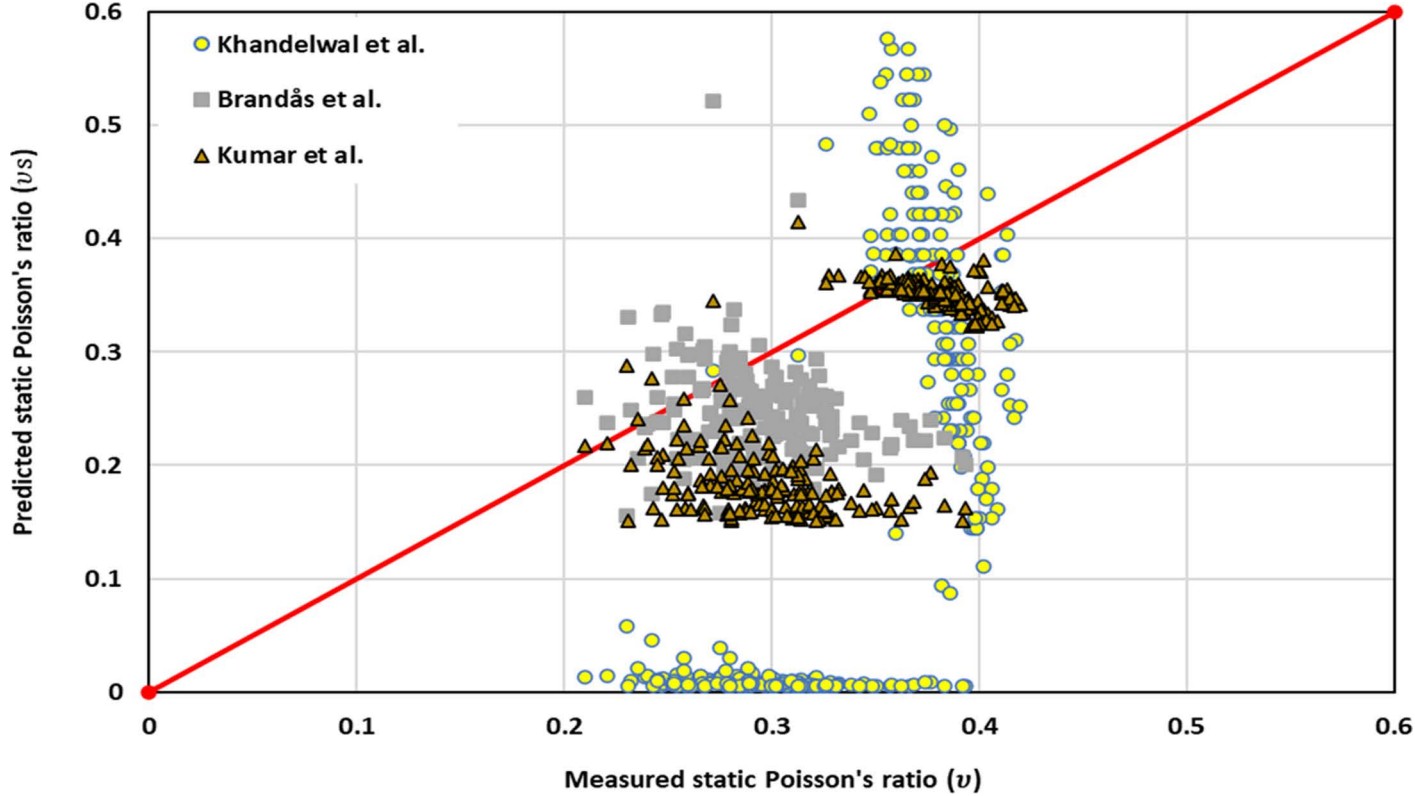

**Fig 20.** Continued.

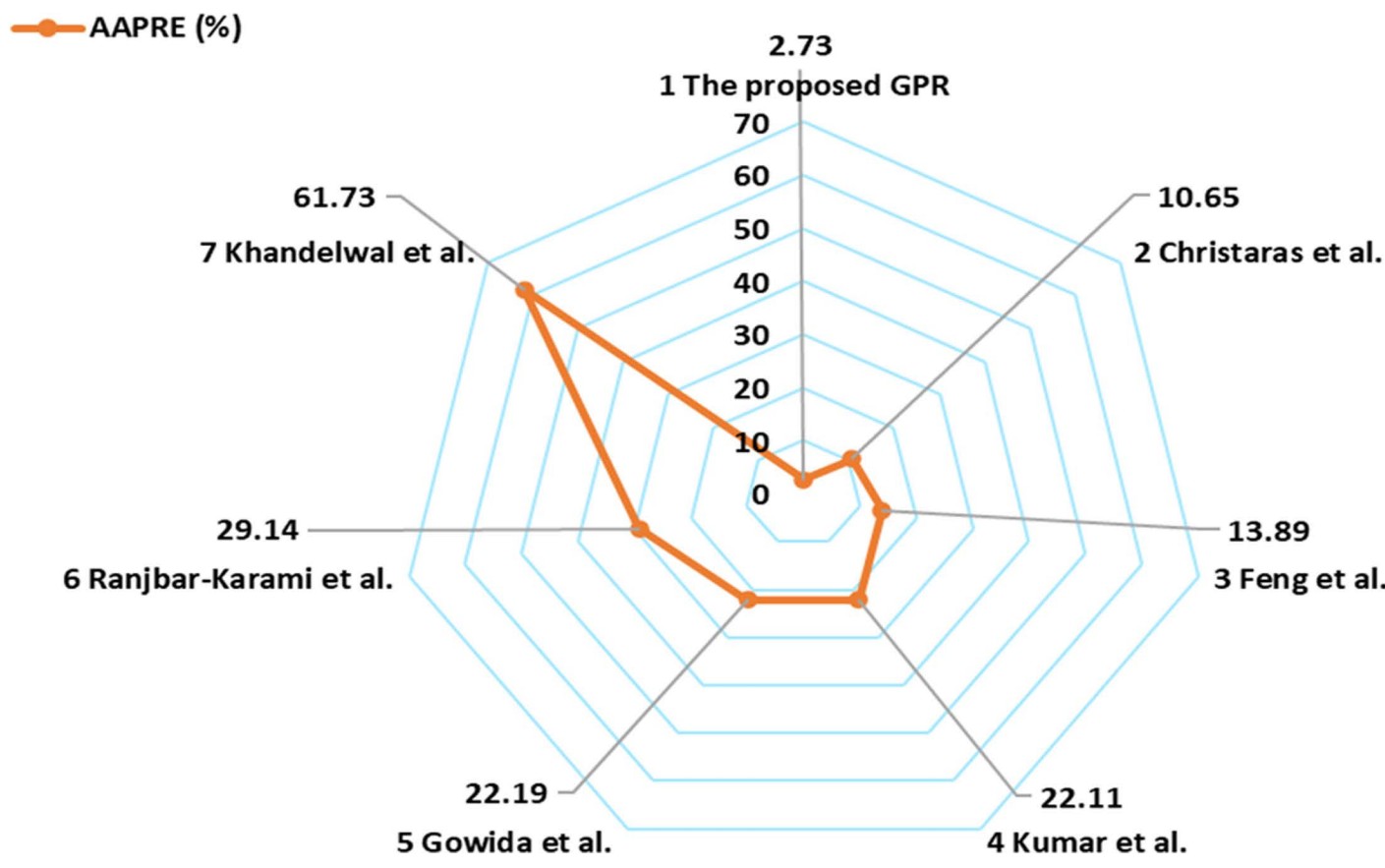

**Fig 21.** (a) Average absolute percentage relative error (AAPRE), and, (b) Coefficient of determination (R²) of the models.

Emax, Emin, SD, and RMSE of (-10 to 110)%(43.9-595)%(0.041-15.6)%, 0.020-520, and 0.1177-1.755, respectively.

**3.3.3 Taylor diagram comparison.** A Taylor diagram is commonly used to summarize the performance of predictive models by comparing their standard deviation and correlation coefficient with a reference dataset (measured dataset). Fig 23 (a, b) illustrates the comparative performance of the different models for predicting $v_s$ to provide a visual representation of the standard deviation and correlation coefficient of each model relative to the reference data. The red arc represents the reference standard deviation, serving as the benchmark against all models which are evaluated. Models closer to the reference curve have a standard deviation that closely matches the observed data.

Kumar et al. [12] model exhibits a close point to the reference curve for the standard deviation but with a lower correlation coefficient of 0.57, Fig 23 (a). Khandelwal et al. [13] model is positioned farther from the reference, with a correlation coefficient of 0.67 and a higher standard deviation of 0.2, Fig 23 (a). Brandås et al. [15] model has a correlation coefficient of 0.77 and a large standard deviation of 0.56, Fig 23 (a). Ranjbar-Karami et al. [14], Christaras et al. [52], Feng et al. [16], and Gowida et al. [17] models show varying levels of performance. Their markers are positioned further from the reference curve, with correlation coefficients of 0.939. The larger spread in standard deviation for the previous models suggests they are less consistent in capturing the variability of the reference data compared to the proposed GPR model.

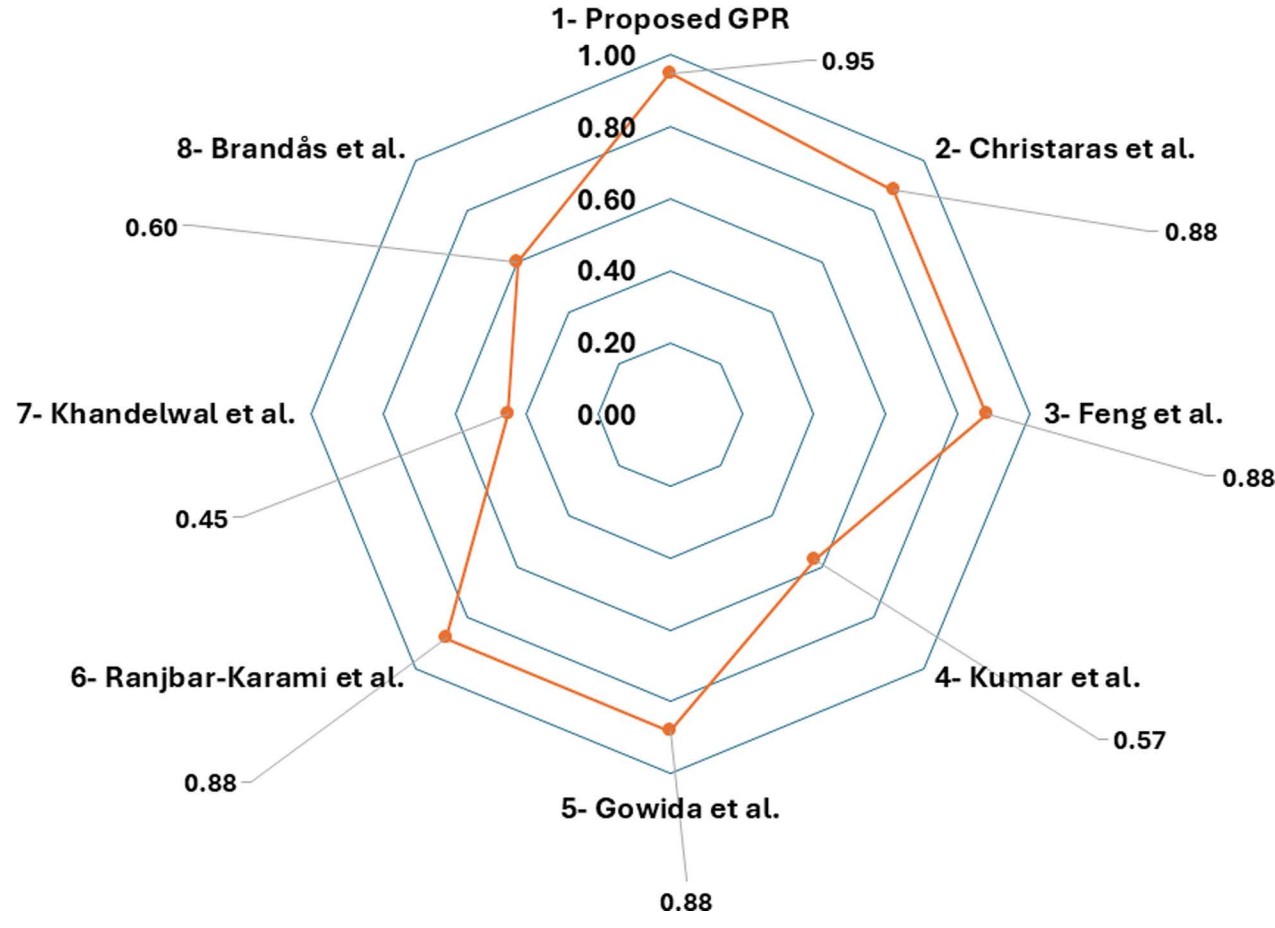

**Fig 21.** Continued.

The proposed GPR model shows superior performance, as evidenced by its proximity to the reference curve and its correlation coefficient of 0.97, Fig 23 (b). This indicates a high degree of accuracy and minimal variance compared to other models. The high correlation coefficient (close to 1) and standard deviation close to the reference for the proposed GPR model signify its robustness and accuracy in modelling $v_s$. The results highlight the superiority of the proposed GPR model in predicting the $v_s$.

**3.3.4 Kruskal-Wallis test comparison.** The Kruskal–Wallis (KW) test evaluates whether there are statistically significant differences among groups. In this study, the Kruskal–Wallis test was taken for the predicted and measured values to $v_s$. The KW test is a nonparametric rank test that tests the null hypothesis that the data from all groups come from a single population. The KW test is a one-way analysis of variance (ANOVA), which assumes a normal distribution of the data. This test can be used in this study to show if there are any differences between the measured and predicted values of the $v_s$ that were obtained from the proposed GPR and previously published models. In this test, the differences between the measured and predicted $v_s$ values can be checked and proved by this test.

The p-values obtained from the statistical analysis represent the probability of observing the data if the null hypothesis (*$H_O$) is true, there is no significant difference between the

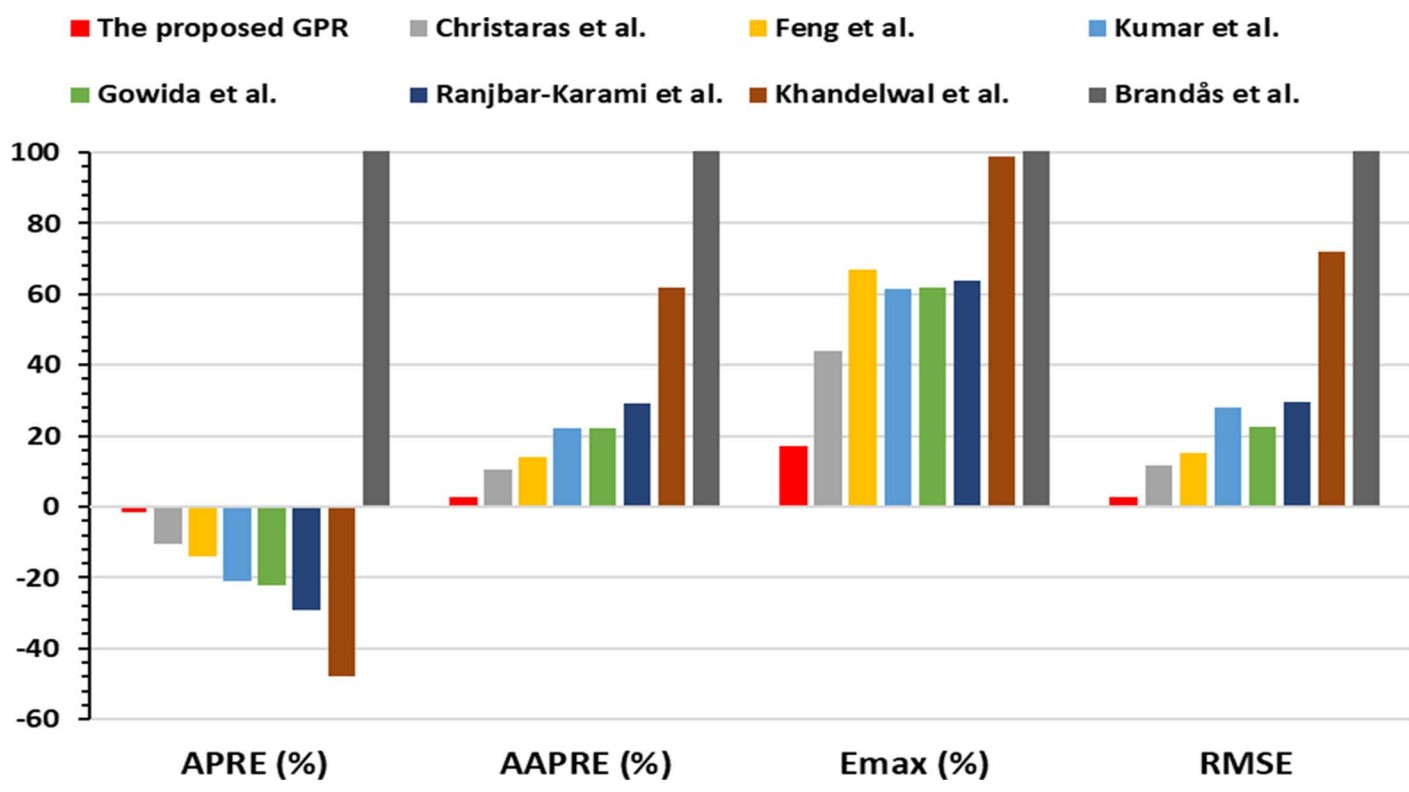

**Fig 22.** (a, b). The statistical error analyses for the models using the same dataset.

measured and predicted values. A p-value greater than 0.05 means failing to reject $*H_O$ or in other words accept the $*H_O$, indicating that the predicted values are not significantly different between the measured and predicted values and, therefore, prove good agreement between the measured and predicted values. Conversely, p-values less than or equal to 0.05 indicate rejecting $*H_O$, proving a significant difference between the predicted and measured values. The p-value serves as a critical metric for assessing the predictive performance of the models [53].

Khandelwal et al. [13], Ranjbar-Karami et al. [14], Brandås et al. [15], Christaras et al. [52], Feng et al. [16], Kumar et al. [12], and Gowida et al. [17] yield extremely low p-values (ranging from $5.828 \times 10^{-34}$ to $2.156 \times 10^{-77}$), Table 4. The null hypothesis is rejected, suggesting these models show statistically significant differences in their predictions compared to the measured dataset. This highlights that these models may not align well with the measured data, potentially indicating suboptimal predictive performance.

With a p-value of 0.808, the null hypothesis is accepted, indicating that the proposed model's predictions show no statistically significant difference between the measured and predicted values, Table 4. This strongly suggests that the GPR model aligns closely with the measured values, demonstrating high predictive accuracy to predict the $\nu_s$.

**3.3.5 Error boxplot and violin graphs comparison.** Fig 24 combines boxplots and violin plots to provide a detailed comparison of errors for the previously published and proposed GPR models. The boxplot, embedded within each violin, highlights key statistical properties such as the median (white dot), interquartile range (IQR, the black rectangle), and whiskers (error ranges within 1.5 times the IQR). These elements summarize the central tendency, variability, and spread of errors. Meanwhile, the violin plot extends this analysis

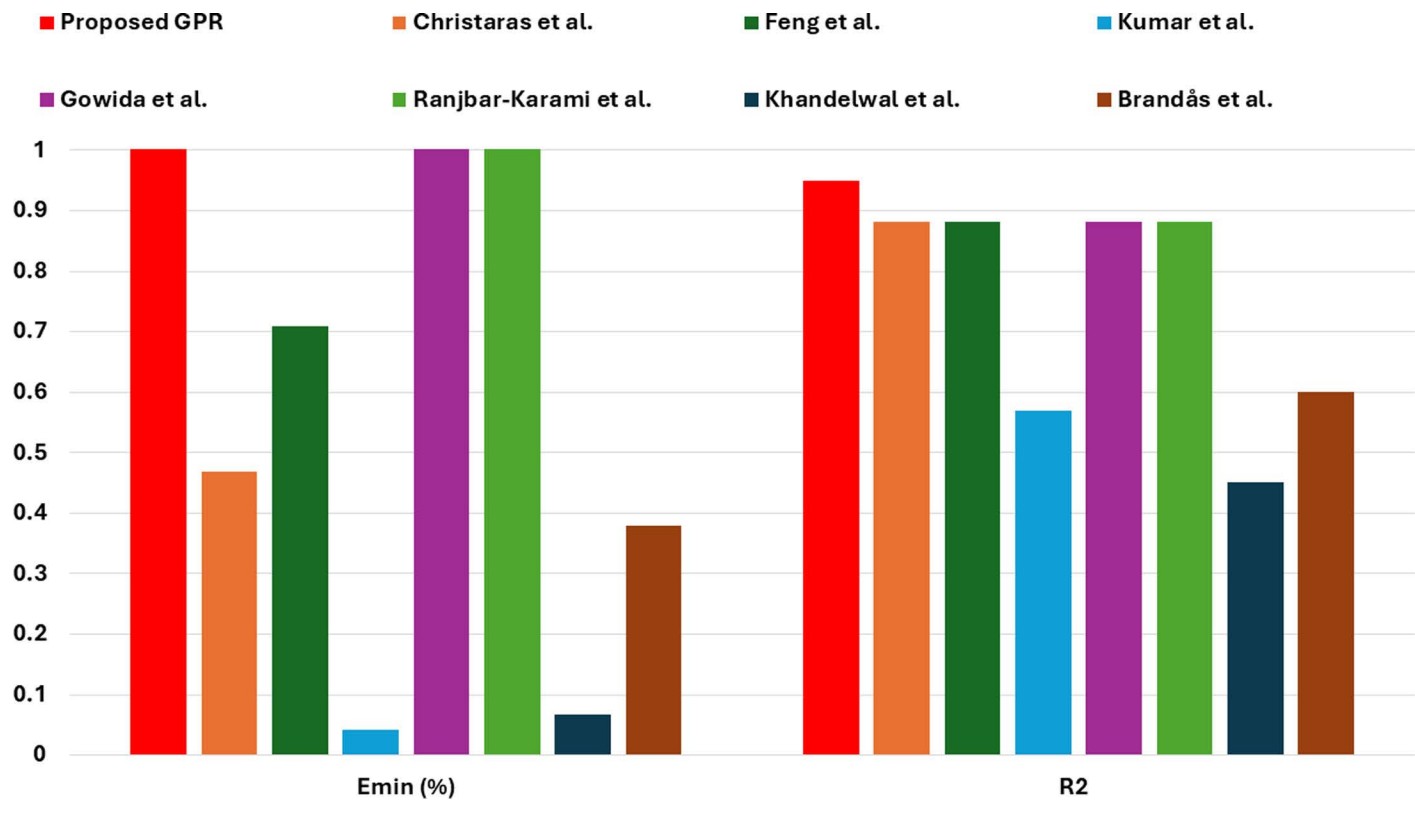

**Fig 22.** Continued.

by showing the distribution and density of errors for the previously published and proposed GPR models. The width of each violin at different error levels reflects the concentration of values, offering insights into error patterns. Together, these plots provide a nuanced understanding of each model's performance, illustrating accuracy. The red horizontal line at 0 serves as a reference for error neutrality, highlighting how well each model's predictions align with the true values.

The proposed GPR model exhibits exceptional performance, as evident from its narrow violin centred around the zero-error line. The median error is almost perfectly aligned with zero, indicating that the model's predictions are accurate. The IQR is very small, meaning most of the errors are tightly concentrated near the median. Additionally, the whiskers are short, and the violin's width is minimal across the entire range, demonstrating that the GPR model has a highly consistent and robust error distribution. This indicates that the model not only minimizes large errors but also avoids producing outliers. The symmetric shape of the violin plot further confirms that the errors are evenly distributed around the median, reinforcing the reliability of the GPR model in accurately predicting the $\nu_s$.

In contrast, the other models exhibit broader and more varied distributions of errors, as shown by their violin and boxplot characteristics. For instance, Brandás et al. [15] model has a significantly wider violin with a negatively skewed distribution, indicating a tendency to underpredict and a lack of consistency. Its median is far from zero, and the IQR and whiskers are wide, signifying larger and more variable errors. Similarly, Ranjbar-Karami et al. [14] model shows a wide violin with a larger spread, although it performs slightly better than Brandás et al. [15] model. Models such as Khandelwal et al. [13], Feng et al. [16], Kumar et

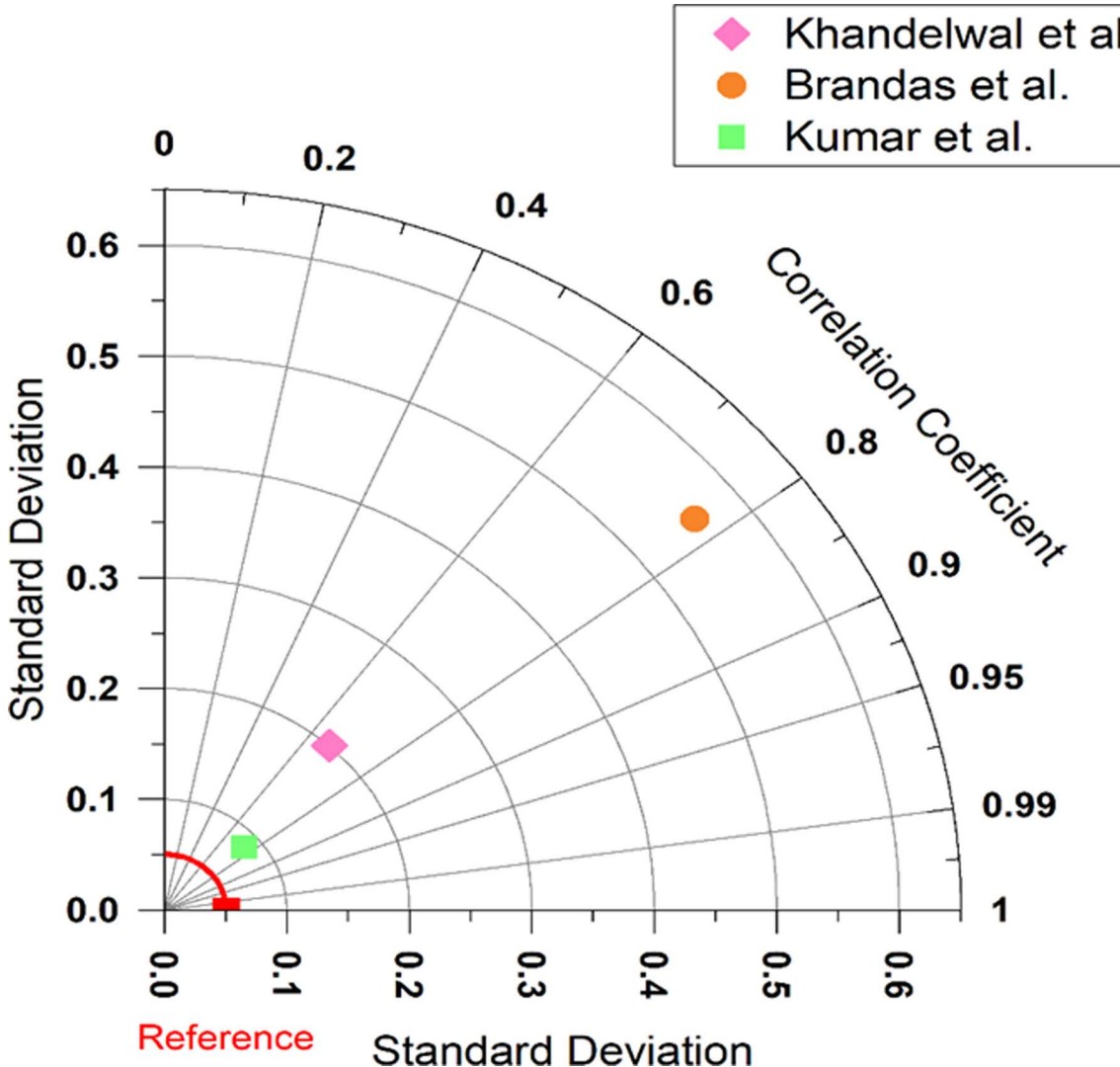

**Fig 23.** (a) Taylor diagrams of Khandelwal et al. [13], Brabdas et al, Kumar et al. [12] models. (b) Taylor diagrams of Ranjbar-Karami et al. [14], Christaras et al. [52], Feng et al. [16], Gowida et al. [17], and the proposed GPR models.

**Table 4. P values of Kruskal–Wallis test at 95% significance level.**

| Model | P-value | *$H_o$ |
|---|---|---|
| Khandelwal et al. [13] | $5.828 \times 10^{-34}$ | reject |
| Ranjbar-Karami et al. [14] | $3.062 \times 10^{-122}$ | reject |
| Brandås et al. [15] | $1.266 \times 10^{-38}$ | reject |
| Christaras et al. [52] | $5.295 \times 10^{-25}$ | reject |
| Feng et al. [16] | $6.471 \times 10^{-29}$ | reject |
| Kumar et al. [12] | $1.672 \times 10^{-31}$ | reject |
| Gowida et al. [17] | $2.156 \times 10^{-77}$ | reject |
| The proposed GPR | 0.808 | **Accept** |

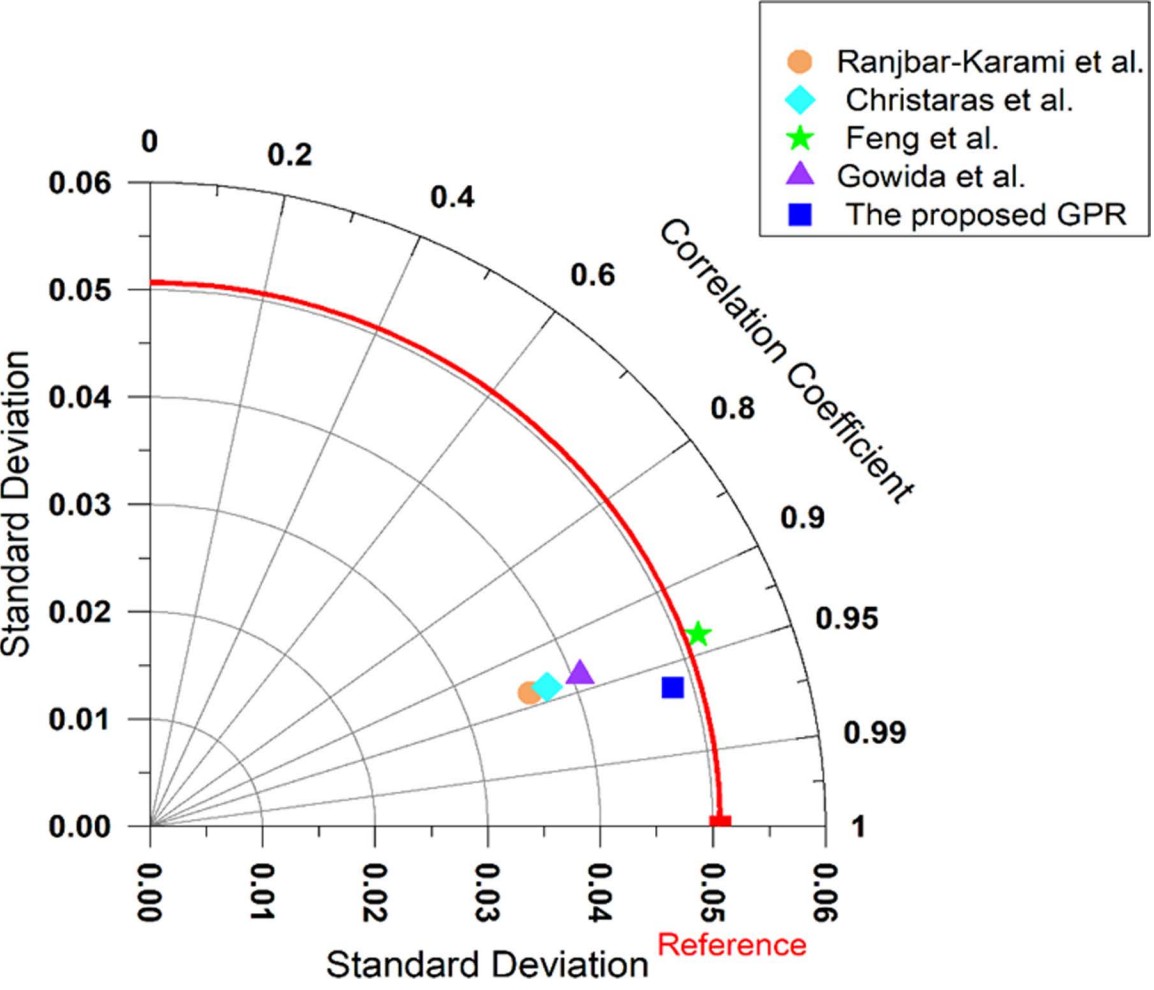

**Fig 23.** Continued.

al. [12], and Gowida et al. [17] show narrower violins and medians closer to zero, indicating moderate accuracy and lower variability. However, their IQRs are still larger compared to the GPR model, and their violins display some asymmetry, suggesting occasional biases or inconsistent predictions. Overall, while some models perform better than others, none match the precision and consistency of the GPR model, as highlighted by the combined boxplot and violin plot analysis.

## 3.4 Results of Poisson's Ratio Application

After the GPR model was developed to predict $v_s$ accurately. The proposed GPR with the previous models were used to determine the fracture pressure. Fig 25 shows the FP based on the previous $v_s$ models. As shown in Fig 25, the FP values were different from the FP values based on the measured or actual $v_s$. However, the FP based on the proposed GPR $v_s$ model has close values to the measured or actual values as shown in Fig 26. As a result, the FP values based on the proposed GPR model show that the proposed GPR model has high accuracy in finding the FP compared to the previous models.

Fig 27 shows the residual error of fracture pressure based on all $v_s$ studied models using the same testing dataset. Most of the previous methods had high residual error due to the

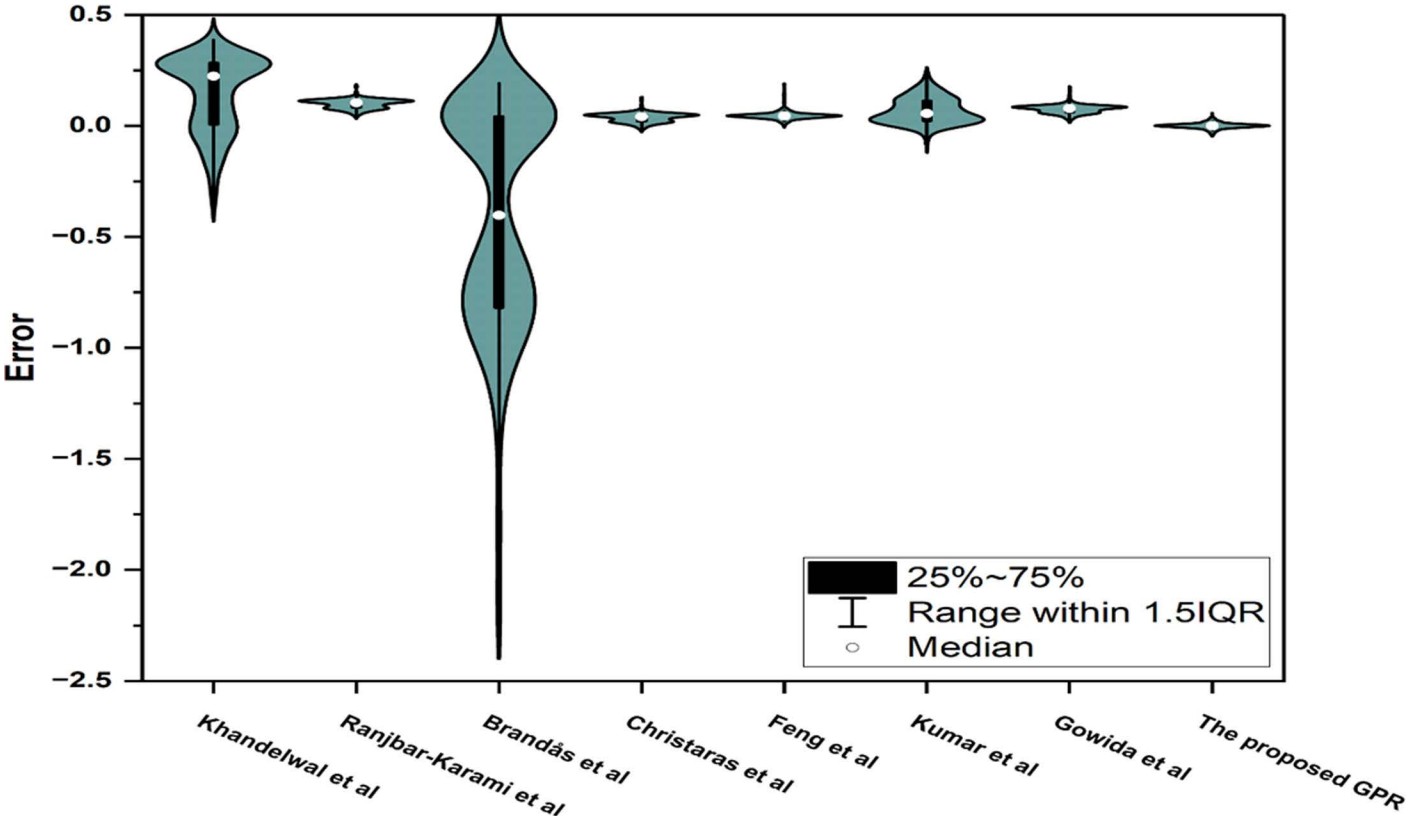

**Fig 24. Error boxplot and violin graphs for the previously published and proposed GPR models.**

$\nu_s$ prediction error as discussed previously. This high residual error of previous models will significantly affect the calculated FP's. The previous models present 87% of the dataset with residual errors (RE) of more than 200 psi, which is unacceptably high. The optimized GPR model reduced the RE of more than 200 psi from 87% to 26%, which is a significant improvement in the reliability of the fracture prediction. Therefore, the improvement of the proposed GPR model's accuracy has a significant effect on the FP calculations.

## 4 Discussion

The results of this paper highlight the superior performance of the optimized GPR model in predicting $\nu_s$ and FP compared to previously published models. The accuracy improvement can be attributed to the fact that the model can learn from a wider range of values, ensuring that it provides more reliable predictions across all ranges of input variables. Ultimately, the inclusion of comprehensive data allows the GPR model to perform better not only in terms of statistical accuracy but also in terms of operational reliability, particularly in critical calculations such as fracture pressure predictions. As a result, the comprehensive dataset from different places, such as the United States, Malaysia, India, Saudi Arabia, and Venezuela contributes to a model that is more accurate, robust, and applicable across a broader spectrum of real-world scenarios. On the other hand, the previous models were created based on the data from specific regions.

The optimized GPR model demonstrated exceptional accuracy, with an $R^2$ value of 0.95 and an AAPRE of 2.73%, which are significantly lower than the AAPRE values of previous models, which ranged from 10.65% to 127.1%. These results indicate that the GPR model not

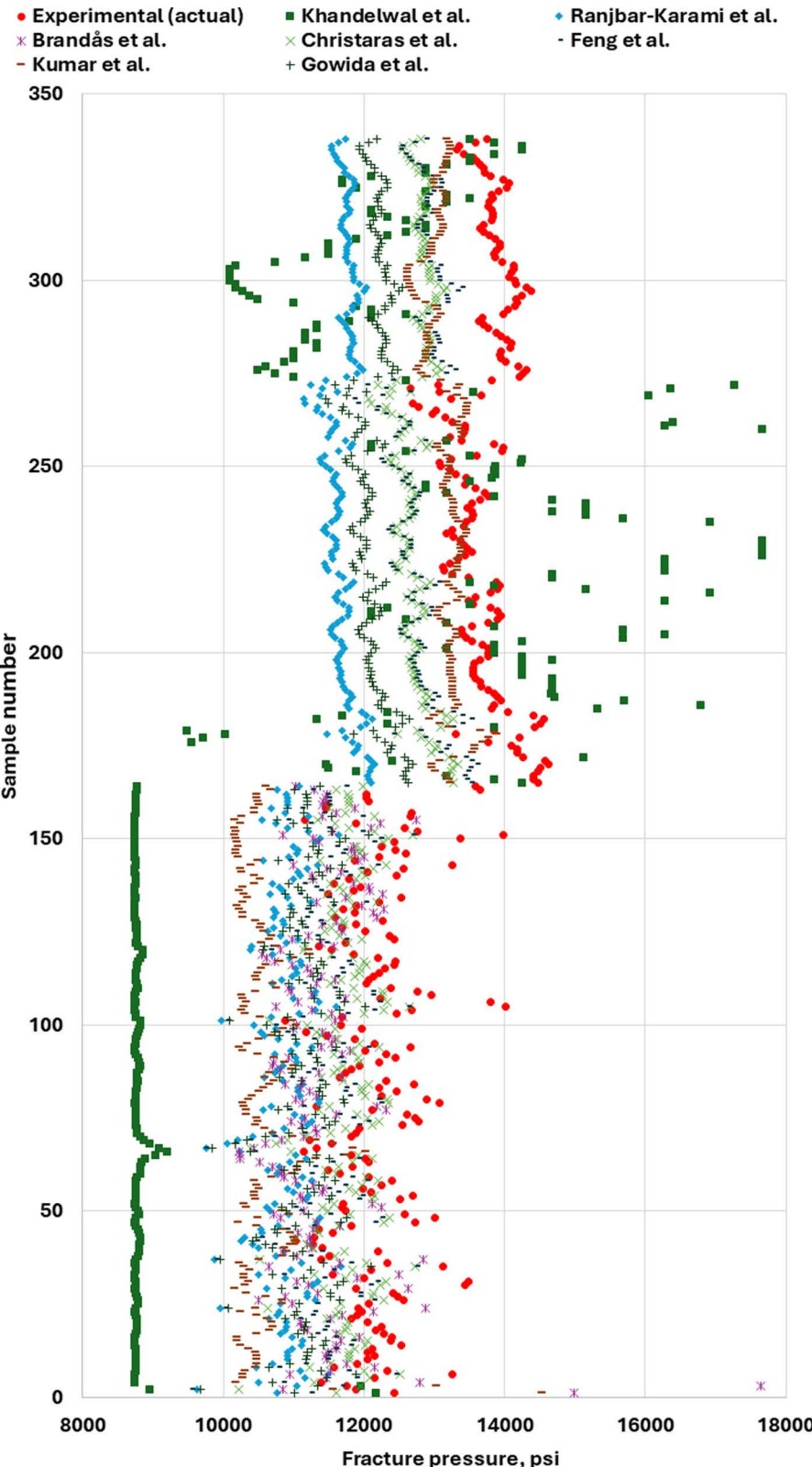

**Fig 25. Fracture pressure based on previous Poisson's ratio models.**

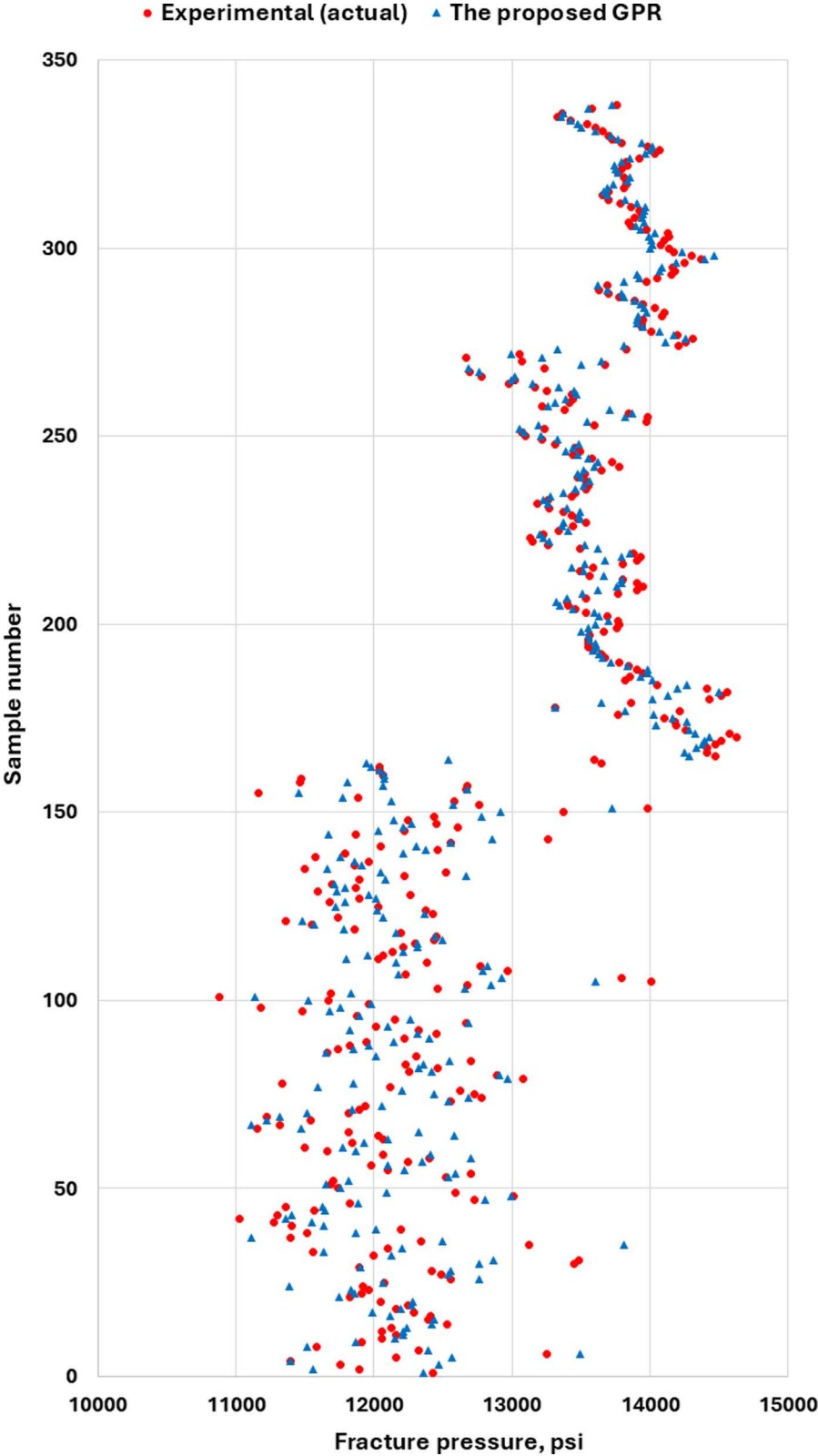

**Fig 26. Fracture pressure based on proposed GPR Poisson's ratio model.**

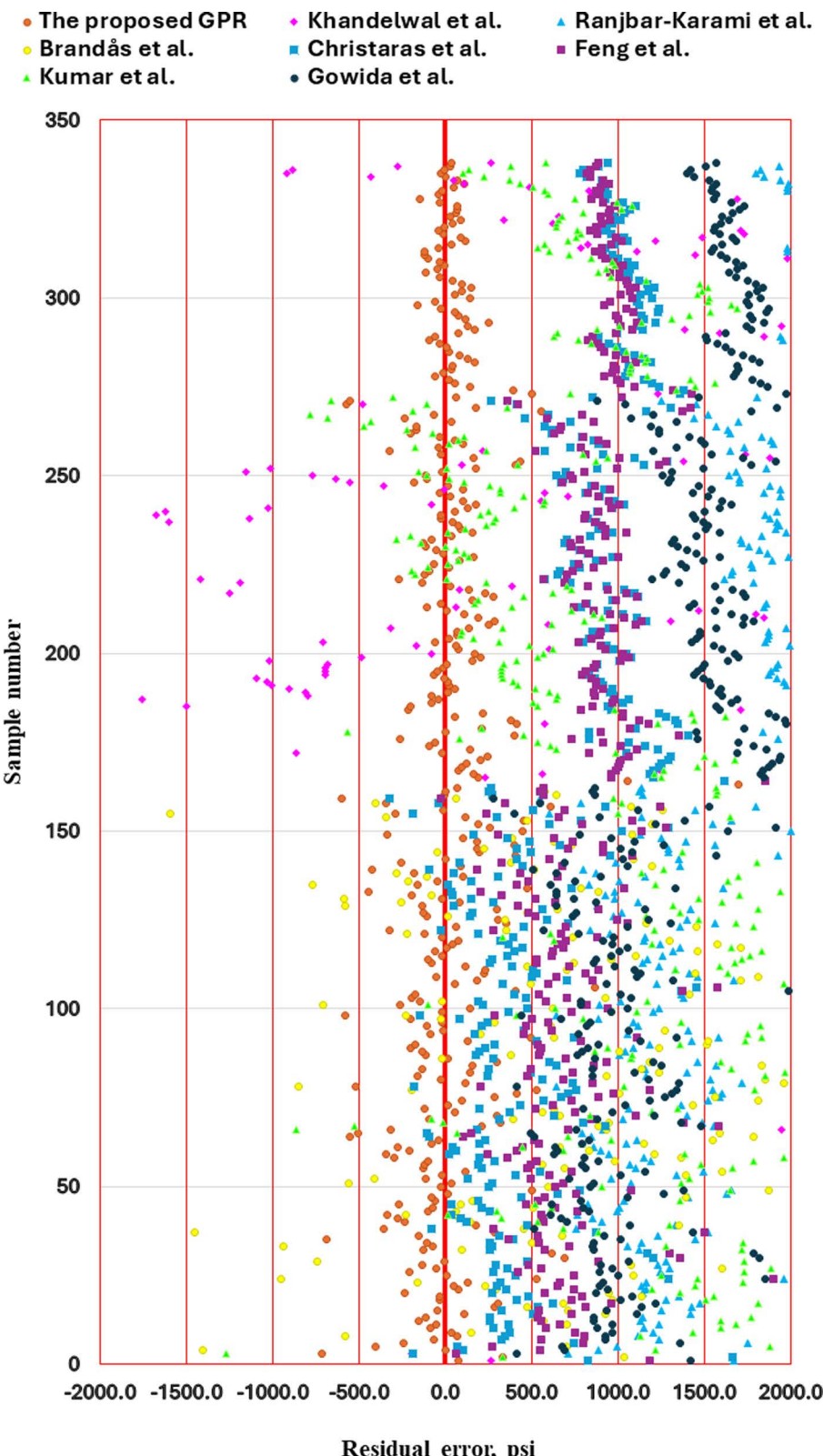

**Fig 27. Residual error of fracture pressure for all studied models.**

only outperforms other models in terms of accuracy but also provides more consistent and reliable predictions across different input ranges. This can be attributed to the GPR model's ability to better capture the complex, nonlinear relationships inherent in the data, making it particularly effective for geomechanical applications. The proposed GPR model demonstrates the correct PB. The TAs of all inputs in the GPR approach validate the accurate relationships. As a result, the proposed GPR model successfully captures the correct PB for all parameters.

Moreover, the GPR model's ability to consistently outperform previous models across multiple evaluation metrics, including RMSE, Emax, and APRE, strengthens its robustness and suitability for practical applications. The GPR model's high accuracy makes it an ideal tool for real-time predictions, offering a more reliable foundation for making informed decisions in wellbore design, fracture pressure management, and reservoir characterization.

The model's consistent performance, even across different input ranges and datasets compared to the previous models, also suggests that it can be generalized and adapted to a wide variety of geomechanical applications, further enhancing its utility in the petroleum industry. The practical implications of the GPR model's performance were demonstrated in the calculation of fracture pressure (FP). When the FP was calculated using $v_s$ predicted by the optimized GPR model, the results were much closer to the actual measured FP values. This is in stark contrast to the previous models, where FP predictions deviated significantly from the measured values, primarily due to inaccuracies in $v_s$ predictions. Notably, the residual error (RE) associated with the FP calculations was reduced from 87% of the dataset showing errors greater than 200 psi in previous models to just 26% in the GPR model. This substantial improvement underscores the critical role of accurate $v_s$ predictions in enhancing the reliability of FP estimations. As FP is a key factor in preventing fractures and blowouts during drilling, the reduction in error can have significant implications for operational safety and efficiency in the petroleum industry

## Conclusions

More than one thousand and six hundred datasets that were publically available were collected from 5 hydrocarbon-producing countries (India, Malaysia, Saudi Arabia, Venezuela, and the United States). After necessary pre-processing to clean the data, the nineteen most common learning methods were used to compute the static poisons ratio ($v_s$), using bulk formation density, compressional time, and shear time as inputs. The performance of these methods was ranked on their low root mean square error (RMSE) and high coefficient of determination ($R^2$). The best-performing ones were selected and further enhanced using various approaches such as trend analysis, group error analyses (GEA), cross-plotting, and statistical error analyses to prove the model's performance and robustness.

The findings of this research are highlighted as follows

- The optimum approach was a Gaussian process regression (GPR) model selected and enhanced out of the 19 current models considered in this study along with eight previously published models. The GPR and current approaches used the same new datasets that were not used to train and evaluate the GPR to show a reasonable comparison.

- The proposed GPR approach had a coefficient of determination ($R^2$) of 0.95 and 2.73% average absolute percentage relative error (AAPRE), which was the lowest AAPRE among the models tested.

- The proposed GPR model was able to depict proper bulk formation density, shear, and compressional times trends as per expectations.

- The proposed GRU approach has the lowest AAPRE and the highest $R^2$ for all datasets: 2% and 0.95 training, 2.87% and 0.94 validation, 2.73% and 0.95 testing, and 2.32% and 0.95 the whole datasets.

- The cross-plotting and group error analysis Figs showed that the optimized GPR approach had high precision for all datasets of various ranges and surpassed the accuracy as compared to other methods considered in this study in all practical ranges.

- The application of $\nu_s$ in determining the fracture pressure (FP) shows that most previous methods had high residual error due to the $\nu_s$ prediction error significantly affecting FP calculations. Consequently, the enhancement of the optimized GPR model's accuracy has a significant effect on the FP.

Machine learning models' accuracy relies on the range, quality, and quantity of the data used for their development. Given the challenges in acquiring data for certain parameters, the proposed models were built using RHOB, DTs, and DTc—commonly utilized parameters in previous studies. The ranges of the RHOB, DTs, DTc, and $\nu_s$ are 0.315-2.994 g/ml, 72.9-341.2 μs/ft, 44.43-186.9 μs/ft, and 0.1627-0.4492, respectively. The advantages of the proposed model far outweigh these shortcomings. The models were constructed using data collected from diverse locations, ensuring extensive data coverage to accurately predict $\nu_s$ across various ranges and regions compared to the previous models. The proposed GPR model underwent comprehensive evaluation through multiple methods, demonstrating its robust predictive capabilities. Trend analysis was conducted to establish appropriate relationships between inputs and outputs, validating the model's alignment with physical behaviour. Additionally, the GPR model was used to predict the fracture pressure accurately.

## Acknowledgements

Deepest gratitude to the Yayasan Universiti Teknologi PETRONAS (YUTP), Universiti Teknologi PETRONAS for supporting this work.

## Author contributions

**Conceptualization:** Fahd Saeed Alakbari, Syed Mohammad Mahmood, Mohammed Abdalla Ayoub, Mysara Eissa Mohyaldinn, Ali Samer Muhsan.

**Data curation:** Fahd Saeed Alakbari, Syed Mohammad Mahmood.

**Formal analysis:** Fahd Saeed Alakbari, Mohammed Abdalla Ayoub, Mysara Eissa Mohyaldinn, Ali Samer Muhsan.

**Funding acquisition:** Syed Mohammad Mahmood, Mohammed Abdalla Ayoub.

**Investigation:** Syed Mohammad Mahmood, Mohammed Abdalla Ayoub, Funsho Afolabi.

**Methodology:** Fahd Saeed Alakbari, Syed Mohammad Mahmood, Mohammed Abdalla Ayoub.

**Project administration:** Syed Mohammad Mahmood, Mohammed Abdalla Ayoub.

**Software:** Fahd Saeed Alakbari, Mohammed Abdalla Ayoub.

**Supervision:** Mohammed Abdalla Ayoub.

**Validation:** Fahd Saeed Alakbari, Syed Mohammad Mahmood, Muhammad Jawad Khan.

**Visualization:** Syed Mohammad Mahmood, Mohammed Abdalla Ayoub.

**Writing – original draft:** Fahd Saeed Alakbari, Syed Mohammad Mahmood, Mohammed Abdalla Ayoub, Muhammad Jawad Khan, Funsho Afolabi.

**Writing – review & editing:** Mysara Eissa Mohyaldinn, Ali Samer Muhsan.

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
