## [Decision Letter · Decision Letter 0]

30 Jul 2024

PONE-D-24-22518Prediction of Poisson’s Ratio for a Petroleum Engineering Application: Machine Learning MethodsPLOS ONE

Dear Dr. Alakbari,

Thank you for submitting your manuscript to PLOS ONE. After careful consideration, we feel that it has merit but does not fully meet PLOS ONE’s publication criteria as it currently stands. Therefore, we invite you to submit a revised version of the manuscript that addresses the points raised during the review process.

**ACADEMIC EDITOR: Major revision**

We look forward to receiving your revised manuscript.

Kind regards,

Agbotiname Lucky Imoize

Academic Editor

PLOS ONE

 [Yayasan Universiti Teknologi PETRONAS (YUTP) (Cost Centre: 015LC0-451)].  

Additional Editor Comments:

The authors should revise the paper according to the two reviewers' comments and edit the English thoroughly.

Reviewers' comments:

Reviewer's Responses to Questions

**Comments to the Author**

1. Is the manuscript technically sound, and do the data support the conclusions?

Reviewer #1: No

Reviewer #2: Yes

2. Has the statistical analysis been performed appropriately and rigorously? 

Reviewer #1: No

Reviewer #2: Yes

3. Have the authors made all data underlying the findings in their manuscript fully available?

Reviewer #1: No

Reviewer #2: Yes

4. Is the manuscript presented in an intelligible fashion and written in standard English?

Reviewer #1: No

Reviewer #2: Yes

5. Review Comments to the Author

Reviewer #1: This manuscript describes an interesting alternative for calculating Statistic Piosson´s ratio based on data-driving models. However, the document in general can be improved. Below are some thecnical comments and suggestions:

1. In section 2.1, please specify whether the datasets were obtained from public sources or not. If not, please clearly state the data sources and provide an assessment of data quality.

2.Between lines 156 and 158, a statement is made that requires support from at least one bibliographic reference. Please check this in other statements in the manuscript.

3 The paragraph beginning at line 166 needs clarification. The key/central idea is not clearly expressed, and the reference to equation (3) is incorrect.

The statement in line 173 is unclear and does not refer to equation (4). This situation also arises between lines 174 and 200.

4. Line 202 incorrectly describes the use of Matlab.

5. Table 1 contains typographical errors.

6. In the paragraph starting at line 227, authors mention using different values of v determined by various methods. Please specify these values and the methods used for that goal.

Reviewer #2: ---Use R2 instead of R.

---Add the MAPE criterion to the table 2.

---Add average-absolute-percentage-relative-error (AAPRE) function.

---Draw a Taylor diagram to compare the methods.

---Compare the measurement and prediction results by performing the Kruskal-Wallis test.

---Draw error boxplot and violin graphs.

---Draw a flow chart explaining the entire work.

---Include the limitations of the study in the conclusion section.

---It needs a good discussion section separate from results.

---Check out the following articles about ANN, GPR and SVM:

Başakın, E. E., Ekmekcioğlu, Ö., Çıtakoğlu, H., & Özger, M. (2022). A new insight to the wind speed forecasting: robust multi-stage ensemble soft computing approach based on pre-processing uncertainty assessment. Neural Computing and Applications, 34(1), 783-812.

Uncuoglu, E., Citakoglu, H., Latifoglu, L., Bayram, S., Laman, M., Ilkentapar, M., & Oner, A. A. (2022). Comparison of neural network, Gaussian regression, support vector machine, long short-term memory, multi-gene genetic programming, and M5 Trees methods for solving civil engineering problems. Applied Soft Computing, 129, 109623.

Citakoglu, H. (2021). Comparison of multiple learning artificial intelligence models for estimation of long-term monthly temperatures in Turkey. Arabian Journal of Geosciences, 14, 1-16.

Citakoglu, H., & Coşkun, Ö. (2022). Comparison of hybrid machine learning methods for the prediction of short-term meteorological droughts of Sakarya Meteorological Station in Turkey. Environmental Science and Pollution Research, 29(50), 75487-75511.

Demir, V., & Citakoglu, H. (2023). Forecasting of solar radiation using different machine learning approaches. Neural Computing and Applications, 35(1), 887-906.

Zouzou, Y., & Citakoglu, H. (2023). General and regional cross-station assessment of machine learning models for estimating reference evapotranspiration. Acta Geophysica, 71(2), 927-947.

6. PLOS authors have the option to publish the peer review history of their article (what does this mean? ). If published, this will include your full peer review and any attached files.

**Do you want your identity to be public for this peer review?** For information about this choice, including consent withdrawal, please see our Privacy Policy .

Reviewer #1: No

Reviewer #2: No

---

## [Author Response · Author response to Decision Letter 1]

27 Dec 2024

Dear respected editor, Dr Agbotiname Lucky Imoize

Academic Editor

PLOS ONE

We greatly appreciate your time and effort in considering & reviewing our paper for your respected journal. We addressed all comments that were given by the reviewers as follows in red colour:

Reviewer #1: This manuscript describes an interesting alternative for calculating Statistic Piosson´s ratio based on data-driving models. However, the document in general can be improved. Below are some technical comments and suggestions:

First, we would like to thank you for considering and giving invaluable comments to improve the manuscript.

1. In section 2.1, please specify whether the datasets were obtained from public sources or not. If not, please clearly state the data sources and provide an assessment of data quality.

Response 1: Thank you for raising this point. The data were collected from different sources in the literature [28–33]. The assessment of data quality was discussed clearly in our previous study [18]. This statement was added to the manuscript accordingly.

2. Between lines 156 and 158, a statement is made that requires support from at least one bibliographic reference. Please check this in other statements in the manuscript.

Response 2. Thank you for your comment. As recommended, between lines (156 and 158 in the previous version) (164-167 new version), two bibliographic references 35 and 36 were added to the statement accordingly.

3 The paragraph beginning at line 166 needs clarification. The key/central idea is not clearly expressed, and the reference to equation (3) is incorrect.

Response 3. Thank you for raising this point. This part was removed from the paper. The paper that discussed this is mentioned, see [48]. The fundamentals and equations for the GPR algorithm were discussed clearly in [48]. This statement was added to the manuscript accordingly.

The statement in line 173 is unclear and does not refer to equation (4). This situation also arises between lines 174 and 200.

Response 3. Thank you for your comment. This part was removed from the paper. The paper that discussed this is mentioned, see [48]. The fundamentals and equations for the GPR algorithm were discussed clearly in [48]. This statement was added to the manuscript accordingly.

4. Line 202 incorrectly describes the use of Matlab.

Response 4. Thank you for raising this point. As suggested, line (202 previous version) (183 new version) was corrected accordingly to describe the use of Matlab.

5. Table 1 contains typographical errors.

Response 5. Thank you for your comment. As recommended, typographical errors were corrected accordingly in Table 1.

6. In the paragraph starting at line 227, authors mention using different values of v determined by various methods. Please specify these values and the methods used for that goal.

Response 6. Thank you for raising this point. As suggested, in the paragraph starting at the line (227 previous version) (226 new version), the values of v which are determined by various methods used for that goal were mentioned accordingly.

Reviewer #2:

First, we would like to thank you for considering and giving invaluable comments to improve the manuscript.

---Use R2 instead of R.

Response 1. Thank you for your comment. As recommended, R2 was used instead of R.

---Add the MAPE criterion to the table 2.

Response 2. Thank you for raising this point. As suggested, the MAPE criterion was added to the table 2.

---Add average-absolute-percentage-relative-error (AAPRE) function.

Response 3. Thank you for your comment. As suggested, the average-absolute-percentage-relative-error (AAPRE) function was added in equation 5 accordingly.

---Draw a Taylor diagram to compare the methods.

Response 4. Thank you for raising this point. As recommended, a Taylor diagram to compare the methods was drawn and discussed in subsection 3.3.3 Taylor Diagram Comparison.

---Compare the measurement and prediction results by performing the Kruskal-Wallis test.

Response 5. Thank you for your comment. As suggested, the measurement and prediction results by performing the Kruskal-Wallis test were compared accordingly in subsection 3.3.4 Kruskal-Wallis Test Comparison.

---Draw error boxplot and violin graphs.

Response 6. Thank you for your suggestion. As recommended, error boxplot and violin graphs were drawn and discussed in subsection 3.3.5 Error Boxplot and Violin Graphs Comparison.

---Draw a flow chart explaining the entire work.

Response 7. Thank you for your comment. As suggested, a flow chart explaining the entire work was drawn and explained accordingly in the first paragraph in section 2 Methodology.

---Include the limitations of the study in the conclusion section.

Response 8. Thank you for raising this point. As recommended, the limitations of the study in the conclusion section were added accordingly.

---It needs a good discussion section separate from results.

Response 9. Thank you for your comment. As suggested, a good discussion section separate from the results was added accordingly in section 4 Discussion.

---Check out the following articles about ANN, GPR and SVM:

Response 10. Thank you for your suggestion. As recommended, the following articles about ANN, GPR and SVM were checked and helped a lot to improve the manuscript.

Başakın, E. E., Ekmekcioğlu, Ö., Çıtakoğlu, H., & Özger, M. (2022). A new insight to the wind speed forecasting: robust multi-stage ensemble soft computing approach based on pre-processing uncertainty assessment. Neural Computing and Applications, 34(1), 783-812.

Uncuoglu, E., Citakoglu, H., Latifoglu, L., Bayram, S., Laman, M., Ilkentapar, M., & Oner, A. A. (2022). Comparison of neural network, Gaussian regression, support vector machine, long short-term memory, multi-gene genetic programming, and M5 Trees methods for solving civil engineering problems. Applied Soft Computing, 129, 109623.

Citakoglu, H. (2021). Comparison of multiple learning artificial intelligence models for estimation of long-term monthly temperatures in Turkey. Arabian Journal of Geosciences, 14, 1-16.

Citakoglu, H., & Coşkun, Ö. (2022). Comparison of hybrid machine learning methods for the prediction of short-term meteorological droughts of Sakarya Meteorological Station in Turkey. Environmental Science and Pollution Research, 29(50), 75487-75511.

Demir, V., & Citakoglu, H. (2023). Forecasting of solar radiation using different machine learning approaches. Neural Computing and Applications, 35(1), 887-906.

Zouzou, Y., & Citakoglu, H. (2023). General and regional cross-station assessment of machine learning models for estimating reference evapotranspiration. Acta Geophysica, 71(2), 927-947.

---

## [Decision Letter · Decision Letter 1]

6 Jan 2025

Prediction of Poisson’s Ratio for a Petroleum Engineering Application: Machine Learning Methods

PONE-D-24-22518R1

Dear Dr. Alakbari,

We’re pleased to inform you that your manuscript has been judged scientifically suitable for publication and will be formally accepted for publication once it meets all outstanding technical requirements.

Kind regards,

Agbotiname Lucky Imoize

Academic Editor

PLOS ONE

Additional Editor Comments (optional):

The revised article is acceptable.

Reviewers' comments:

Reviewer's Responses to Questions

**Comments to the Author**

1. If the authors have adequately addressed your comments raised in a previous round of review and you feel that this manuscript is now acceptable for publication, you may indicate that here to bypass the “Comments to the Author” section, enter your conflict of interest statement in the “Confidential to Editor” section, and submit your "Accept" recommendation.

Reviewer #1: All comments have been addressed

2. Is the manuscript technically sound, and do the data support the conclusions?

Reviewer #1: Partly

3. Has the statistical analysis been performed appropriately and rigorously? 

Reviewer #1: Yes

4. Have the authors made all data underlying the findings in their manuscript fully available?

Reviewer #1: Yes

5. Is the manuscript presented in an intelligible fashion and written in standard English?

Reviewer #1: No

6. Review Comments to the Author

Reviewer #1: The manuscript has been revised and now aligns well with scientific standards and interests on the domain. It presents potentially valuable results that could have significant applications in the industry.

7. PLOS authors have the option to publish the peer review history of their article (what does this mean? ). If published, this will include your full peer review and any attached files.

**Do you want your identity to be public for this peer review?** For information about this choice, including consent withdrawal, please see our Privacy Policy .

Reviewer #1: **Yes: ** Víctor Flores

---

## [Editor Report · Acceptance letter]

PONE-D-24-22518R1

PLOS ONE

Dear Dr. Alakbari,

I'm pleased to inform you that your manuscript has been deemed suitable for publication in PLOS ONE. Congratulations! Your manuscript is now being handed over to our production team.

Kind regards,

on behalf of

Mr. Agbotiname Lucky Imoize

Academic Editor

PLOS ONE